# Resource-aware DBSCAN-based re-clustering in hybrid C-V2X/DSRC vehicular networks

**Jaafar Sadiq Alrubaye**[1,2], **Behrouz Shahgholi Ghahfarokhi**[1] *

1 Faculty of Computer Engineering, University of Isfahan, Isfahan, Iran, 2 Faculty of Computer Science, University of Wasit, Wasit, Iraq

* shahgholi@eng.ui.ac.ir

## Abstract

5G wireless networks are paying increasing attention to Vehicle to Everything (V2X) communications as the number of autonomous vehicles rises. In V2X applications, a number of demanding criteria such as latency, stability, and resource availability have emerged. Due to limited licensed radio resources in 5G cellular networks, Cellular V2X (C-V2X) faces challenges in serving a large number of cars and managing their network access. A reason is the unbalanced load of serving Base Stations (BSs) that makes it difficult to manage the resources of the BSs optimally regarding the frequency reuse in cells and its subsequent co-channel interference. It is while the routing protocols could help redirect the load of loaded BSs to neighboring ones. In this article, we propose a resource-aware routing protocol to mitigate this challenge. In this regard, a hybrid C-V2X/ Dedicated Short Range Communication (DSRC) vehicular network is considered. We employ cluster-based routing that enables many cars to interface with the network via some Cluster Heads (CH) using DSRC resources while the CHs send their traffic across C-V2X links to the BSs. Traditional cluster-based routings do not attend the resource availability in BSs that are supporting the clusters. Thus, our study describes an enhanced clustering method based on Density-Based Spatial Clustering of Applications with Noise (DBSCAN) that re-clusters the vehicles based on the resource availability of BSs. Simulation results show that the proposed re-clustering method improves the spectrum efficiency by at least 79%, packet delivery ratio by at least 5%, and load balance of BSs by at least 90% compared to the baseline.

## Introduction

In the era of the Internet of Things (IoT), Vehicle to Everything (V2X) connectivity is a new paradigm. V2X communication can be traced back to Vehicular Ad hoc Networks (VANETs). The standard presented by Institute of Electrical and Electronics Engineers for vehicular networks (IEEE 802.11p) and the Dedicated Short-Range Communication (DSRC)/Wireless Access in Vehicular Environment (WAVE) standard are the original communication technologies for VANETs that offer wireless connectivity between vehicles and between vehicles and roadside infrastructure [1, 2].

**Data Availability Statement:** No dataset was created during this research. We use the dataset introduced in Ref. [40] which is accessible from

https://www.fhwa.dot.gov/publications/research/operations/07030/.

**Funding:** The author(s) received no specific funding for this work.

**Competing interests:** The authors have declared that no competing interests exist.

DRSC is a wireless communication technology that allows for short-range communication between vehicles, other road users (pedestrians, cyclists, etc.), and roadside infrastructure (traffic signals, electronic message signs, etc.). DSRC usually operates in the 5.9 GHz band and exploits IEEE 1609.3, IEEE 1609.4, and IEEE 802.11p protocols in various layers of the protocol stack [3]. On the other hand, Cellular-V2X (C-V2X) is a recent alternative to classic IEEE 802.11p and DSRC/WAVE technologies. 3$^{rd}$ Generation Partnership Project (3GPP) recommends Long-Term Evolution V2X (LTE-V2X) services for transportation systems in Release 14. It has standardized two new sidelink transmission modes for LTE-V2X, i.e., sidelink mode 3 and mode 4. In sidelink mode 3, one or more synchronous evolved NodeBs (eNBs) has/have the responsibility for resource scheduling and interference management of all the LTE-V2X links. In mode 4, all the scheduling and management mechanisms are done by vehicles. LTE-V2X provides higher bandwidth, higher transmission rates, and larger coverage area compared to DSRC. LTE-V2X reuses existing cellular infrastructure and spectrum, allowing for real-time vehicular communications. It is also used for non-safety applications, such as traffic information transmission [3, 4]. Moreover, 5$^{th}$ Generation (5G) New Radio (NR) that is standardized by 3GPP offers connectivity for V2X. In 5G NR, for V2X communication, two interfaces are designed, which are used in Vehicle to Network (V2N), Vehicle to Infrastructure (V2I), Vehicle to Pedestrian (V2P), and Vehicle to Vehicle (V2V) communications [5].

Accordingly, C-V2X facilitates communications over greater distances than DSRC/WAVE and IEEE 802.11p, even though C-V2X presents challenges in terms of resource management. Different Resource Allocation (RA) strategies are presented for C-V2X networks [6] that allow both Vehicle to Infrastructure (V2I) and V2V communications. V2I systems need to quickly respond to self-driving cars. During busy times or emergencies, when there are a lot of requests at a Base Station (BS) or what we called eNodeB (eNB) in LTE, it becomes difficult for V2I systems to keep working well, which can be dangerous. The study of [7] aims to improve V2I communication under uncertain request arrivals. To reach this goal, they suggest a communication system with User Equipment (UE), Road Side Units (RSUs), and BSs, and apply various resource management techniques to achieve high reliability requirements. Authors of [8] suggest a step-by-step approach to make V2I communications more efficient by considering how the interference between directional beams affects capacity. They also think about a simple resource allocation plan that is much easier to compute than the repetitive plan and doesn't really affect performance. A similar challenge also exists for V2V communications. Ref. [9] utilizes both fine and coarse geo knowledge for V2V Resource Allocation (RA) where the area is divided into small or big sub-areas and Resource Blocks (RBs) are considered for each sub-area. Vehicles use an energy-sensing technique to select the appropriate RB from those reserved for that sub-area. However, Ref. [10] uses a map-based strategy for V2V RA that does not require energy monitoring. Methods such as [11] employ clustering and allocate resources to each Cluster Head (CH) to assign them to the cluster members for V2V communications. The authors propose two heuristic algorithms in [12] that enable the use of some RBs already assigned to V2V communications for clustered V2I communications without significantly affecting Quality of Service (QoS) requirements of V2V links.

Despite the many proposed resource management methods, C-V2X still continues to face the challenge of limited radio resources especially for V2I communications. A reason is the unbalanced load of serving BSs in V2I communications that makes it difficult to manage the reuse of radio resources in cells appropriately regarding the subsequent co-channel interference. This challenge is more elaborated in 5G by employing ultra-dense networks and small cells. This is while routing algorithms could help in resource management if those were resource-aware and could redirect the load of saturated BSs to neighboring ones based on resource availability knowledge to allow better V2I radio resource management.

In classic VANETs, routing protocols were utilized to facilitate long-distance communications via multi-hop transmission. VANETs utilize numerous routing protocols, including position-based [13], topology-based [14], broadcast-based [15, 16] and cluster-based routing [17–20]. Proactive, reactive, and multipath routing algorithms are compared using simulations in research [21]. In particular, the effectiveness of the Ad Hoc On-Demand Distance Vector (AODV), Destination Sequenced Distance Vector (DSDV), and Ad-hoc On-demand Multipath Distance Vector (AOMDV) protocols is assessed regarding the number of nodes, pause times, and number of traffic connections. Owing to frequent shift in movement patterns, a fuzzy method is proposed to ensure stability in route creation and to prioritize emergency packets in AODV [22]. In [23] the Optimized Link State Routing Protocol (OLSR) performance is evaluated with respect to various random and group mobility models. However, cluster-based routing protocols offer higher benefits in V2X [24]. They adopt car clustering with the use of position based, hierarchical based, and density-based clustering techniques such as k-means and Density Based Spatial Clustering of Applications with Noise (DBSCAN) algorithms. DBSCAN has several advantages over other clustering algorithms. DBSCAN is robust to outliers (noise points). This means that it can effectively identify and separate noise points from the clusters, which can improve the quality of the clustering result. It also can effectively cluster data even into complex shapes, which is a challenge of other clustering algorithms. It does not need to specify the number of clusters a priori, unlike many other clustering algorithms [24, 25]. Using cluster-based routing in hybrid C-V2X/DSRC networks, the traffic is first routed to cluster heads using DSRC links, and then it is transferred to the core network using C-V2X links. Nevertheless, past techniques of resource allocation in C-V2X did not consider routing capabilities in their solution. On the other hand, previous approaches to cluster-based routing did not take into account the availability of cellular radio resources when they are clustering the vehicles.

This research examines the idea of resource-aware cluster-based routing in heterogeneous DSRC/C-V2X networks intending to improve spectrum efficiency. To this aim, we assume clusters of vehicles where their traffic is sent to CHs using DSRC radio resources and then forwarded to the core network using cellular radio resources. Our proposed approach attempts to balance the load of BSs via a resource-aware re-clustering algorithm regarding the resource availability of the BSs that are serving CHs. The proposed re-clustering algorithm alters DBSCAN algorithm so that it takes the radio resource availability of BSs into account. Briefly, the contributions of this paper are as follows:

- Discussing necessity of resource-awareness in VANET routing and explaining some routing scenarios that resource-awareness may improve their performance in C-V2X networks.

- Bringing up the resource-awareness in one of the possible routing scenarios, i.e., cluster-based V2I communications.

- Proposing a resource-aware re-clustering method based on DBSCAN algorithm for C-V2X/DSRC networks to improve spectrum efficiency and load balance of BSs in mentioned scenario.

Table 1 shows the abbreviation used in this paper. The following is how the structure of this document is set up. Relevant work will be reviewed in next section. Resource-aware routing paradigm is discussed after Related Work section. The explanation of the system model and the proposed solution can be found in section called Proposed Method. The simulation findings are presented after that, followed by a discussion of the conclusion and an examination of prospective future work in last section.

**Table 1. Abbreviations.**

| Abbreviation | Description | Abbreviation | Description |
|---|---|---|---|
| 3GPP | 3rd Generation Partnership Project | IEEE | Institute of Electrical and Electronics Engineers |
| 5G NR | 5th Generation New Radio | LTE | Long Term Evolution |
| ANFIS | Adaptive Neuro Fuzzy Inference System | M2M | Machine to Machine |
| AODV | Ad-hoc On-Demand Distance Vector | NOMA | Non-Orthogonal Multiple Access |
| AOMDV | Ad-hoc On-demand Multipath Distance Vector | OLSR | Optimized Link State Routing Protocol |
| BS | Base Station | QoS | Quality of Service |
| CDS | Cooperative Driving System | RB | Resource Block |
| CH | Cluster Head | RN | Relay Node |
| CM | Cluster Member | RSS | Received Signal Strength |
| C-V2X | Cellular-Vehicle to Everything | RSU | Road Side Unit |
| D2D | Device to Device | V2N | Vehicle to Network |
| DBSCAN | Density-Based Spatial Clustering of Applications with Noise | V2P | Vehicle to Pedestrian |
| DSDV | Destination Sequenced Distance Vector | V2V | Vehicle to Vehicle |
| DSRC | Dedicated Short Range Communication | C-V2X | Vehicle to Everything |
| FLC | Fuzzy Logic Control | VANET | Vehicular Ad hoc Networks |
| FV | Free Vehicle | | |

## Related work

As previously stated, cellular networks have provided greater performance to enable connected automobiles in recent years due to many weaknesses in traditional VANET technology. Moreover, C-V2X is distinguished by the fact that cars can connect directly to the BS, establish connections with other vehicles, and transfer vehicle information at high data rates using cellular radio resources [5]. However, due to traffic congestion, there are insufficient cellular radio resources. To maintain the network's viability, managing resources in high-density areas has proven to be very difficult. Thus, this section discusses recent research on the aforementioned C-V2X challenge. This section also examines the recent works on routing protocols for VANETs.

Reference [26] presents a centralized solution to resource allocation employing Non-Orthogonal Multiple Access (NOMA) by dividing cars into multiple groups and allocating resources to each group based on the placements of its members. The purpose of this work is to optimize performance based on the delivery ratio of packets. Alternatively, Ref. [11] presents a cluster-based method for resource allocation in which vehicles may be CH, Cluster Member (CM), or Free Vehicle (FV). Upon reaching a predefined energy threshold, the vehicle will apply to join the cluster. In this scenario, energy-sensing algorithms are used to create clusters, but there is no consideration given to the capability of the BS to support clusters of uncontrolled size regarding resource limitations. Similarly, Ref. [9] presents a geo-based approach for resource allocation by dividing the area into predetermined-size sub-areas. The mapping specifies the RBs allotted to a specific region. This map is provided for all automobiles in the region. This map is announced to all regional vehicles. Ref. [27] presents a method for selecting the optimal RSU during handoff. Once authenticated, RSU allocates resources to the vehicles using deep Q-learning algorithm. In Ref. [28], authors use the Adaptive Neuro Fuzzy Inference System (ANFIS) to solve the foregoing issue of allocating resources to prominent Machine to Machine (M2M) devices. The implementation of rules in ANFIS will entail the distribution of resources beginning with the device with the highest priority. A learning-based resource selection (decentralized RB allocation) is illustrated in [29] where the authors give a deep reinforcement learning algorithm for optimizing resource allocation. When the

load of sub-areas varies, interference control is the most difficult component of this task. Also, limited resources aren't considered in this paper.

Traditional VANETs make use of DSRC or IEEE 802.11p for short-range communications. However, packet routings are required for long-distance communications to be provided [30]. Ref. [31] outlines a cluster-based routing system that can boost the network's scalability by dividing neighboring vehicles into clusters. The DBSCAN approach is utilized to construct clusters based on location data. There, message delivery might be prioritized for densely populated areas. Because urgent message transmission is the main element of VANETs, accompanying issues, such as high mobility, poor connections, and the dynamic nature of vehicles, must be tackled dynamically. More to add, obtaining information about the high-density and low-density regions can also aid in avoiding the problems associated with sparse VANETs. Pipelining the greedy forward method and clustering algorithm improves the performance of broadcasting messages over the VANET and prioritizes the transmission of urgent messages over the VANET [32]. Ref. [33] assesses VANET routing systems in terms of scalability, dependability, resource scarcity, and the hidden terminal problem. The lack of gateway devices in the VANET's flat V2V network layout might lead to problems with scalability, resource scarcity, dependability, and concealed terminal difficulties. In order to handle all of these problems and enhance network performance, the idea of vehicle clustering invented. Based on the protocols' design goals, Ref. [33] offers a detailed categorization of clustering algorithms in VANET. In [34], context-based and geographical grouping techniques are combined. In addition, destination-aware routing protocol which decreases end-to-end delay and increases packet delivery ratio is suggested for inter-clustering routing. Ref. [35] proposes a unique stable clustering technique utilizing DBSCAN in the V2V region to ensure a steady live road surveillance service with no disruptions for vehicles with insufficient visual area. The authors use DBSCAN to construct clusters and employ fuzzy logic to choose the ideal cluster head. To prevent a phenomenon known as "broadcast storm", which frequently occurs in VANETs and causes the majority of collisions, Ref. [36] proposes a logic-based strategy for VANET maintenance and improvement using DBSCAN to develop a customized algorithm for cluster generation in a centralized manner. However, the above methods do not take the knowledge about radio resource availability into account. A modification to the distributed scheduling of LTE-V sideline mode 4 for 5G V2X communications is recommended in 3GPP. But using cellular resources for V2V communications has the disadvantage of having restricted radio resources [5]. DBSCAN introduced to the Cooperative Driving System (CDS) based on a 5G network architecture with a resource allocation technique inspired by Device-to-Device (D2D) communications [37]. The proposed network design and cooperative behavior-based method contribute to improving CDS QoS. The vehicular clustering system that uses DBSCAN algorithm demonstrates a significant boost in throughput but does not yet effectively account for resource availability.

To the best of our knowledge, earlier methods for improving the utilization of restricted cellular radio resources in C-V2X do not incorporate routing capabilities. On the other side, current cluster-based routing methods are not resource-aware. Therefore, this study proposes a resource-aware clustered routing for hybrid C-V2X/DSRC networks to enhance the utilization of cellular radio resources. Table 2 summarizes the top related work and their disadvantages in brief. As seen, a drawback of previous works is lack of attending the resource availability in routing decisions, and specially in cluster formation.

## Resource-aware routing in C-V2X

As mentioned before, radio resources management is an important challenge in C-V2X networks and resource-aware routing can help in better management of radio resources.

**Table 2. Summary of related work.**

| Ref. | Technology | Resource Allocation | Routing | Clustering | Details | Disadvantages |
|---|---|---|---|---|---|---|
| [5] | C-V2V | √ | - | - | Improving V2V communications to autonomously select their radio resources using a distributed scheduling scheme supported by congestion control mechanisms. | • Is not routing-aware<br>• Does not attend to V2I links |
| [4] | C-V2I | √ | - | - | Handoff decision and access network selection using game theory | • Is not routing-aware |
| [7] | C-V2X | √ | - | - | Resource management techniques to minimizes the shadowing area in the 5G-NR V2X environment, and to maximize the average throughput of vehicles | • BSs' loads are not considered in high density areas |
| [9] | C-V2X | √ | - | - | Geo-based scheduling scheme for autonomous radio resource selection | • Is not routing-aware<br>• Does not balance the loads of BSs |
| [11] | C-V2X | √ | - | √ | Cluster-based resource selection scheme to reduce the resource collision | • Is not routing-aware<br>• Does not balance the loads of BSs |
| [12] | C-V2X | √ | √ | √ | Geo based clustering scheme that enables use of some RBs already assigned to V2V communications for clustered V2I communications with lower wastage | • Does not balance the loads of BSs |
| [26] | C-V2X | √ | - | - | Using energy sensing and NOMA to allow groups of vehicles share the same sub-resources | • Is not routing-aware<br>• Does not balance the loads of BSs |
| [28] | C-V2X | √ | - | - | Uses ANFIS to make priority for massive M2M communications | • Does not attend resource availability of BSs |
| [27] | C-V2I | √ | - | - | A method for selecting the optimal RSU during handoff | • Does not attend resource availability of BSs |
| [29] | C-V2V | √ | - | - | Decentralized resource allocation mechanism for V2V communications using deep reinforcement learning | • Does not consider spectrum efficiency<br>• Does not attend resource availability of BSs |
| [30] | C-V2X | √ | - | √ | Two-level clustering scheme for efficient data dissemination in C-V2X | • Does not attend resource availability |
| [34] | V2V | √ | √ | √ | Hybrid clustering and destination-aware routing to reduces the control overhead and end-to-end delay while improving the overall packet delivery ratio | • Does not attend resource availability |
| [35] | V2V | - | √ | √ | Integrating DBSCAN with Fuzzy Logic Control (FLC) to form clusters with best cluster head. | • Is not resource aware |
| [37] | C-V2X | √ | - | √ | Cooperative behavior-based scheme for improving QoS for CDS by incorporating DBSCAN | • Does not consider current available resources |
| Proposed | Hybrid C-V2X/DSRC | √ | √ | √ | Managing resources using re-clustering based on DBSCAN algorithm to achieve better load balancing | • Does not pay attention to the capacity of DSRC in supporting cluster members. |

Resource-aware routing is a powerful approach that may improve the performance and efficiency by creating the paths or clusters based on the radio resource availability or modifying them towards improving the spectrum efficiency or load balance of cells. Various routing scenarios are possible where resource-awareness can be considered there. Routing protocols are used in V2V communications to form multi-hop links between source and destination vehicles. This type of routing uses radio resources in a multi-hop manner both for unicast and multicast/geocast communications. Resource-awareness is essential in this type of routing, both in geo-based [9] and map-based [10] techniques, to better manage the routes with respect to the available resources in various geographical locations.

Furthermore, routing is used in V2I scenarios for communications from/to the BS for various applications such as traffic management, Internet access, and entertainment. V2I routing is performed in cluster-based or multi-hop manner. In cluster-based V2I routing, traffic is routed to/from the BS through a cluster head [12] while in multi-hop routing, the traffic is

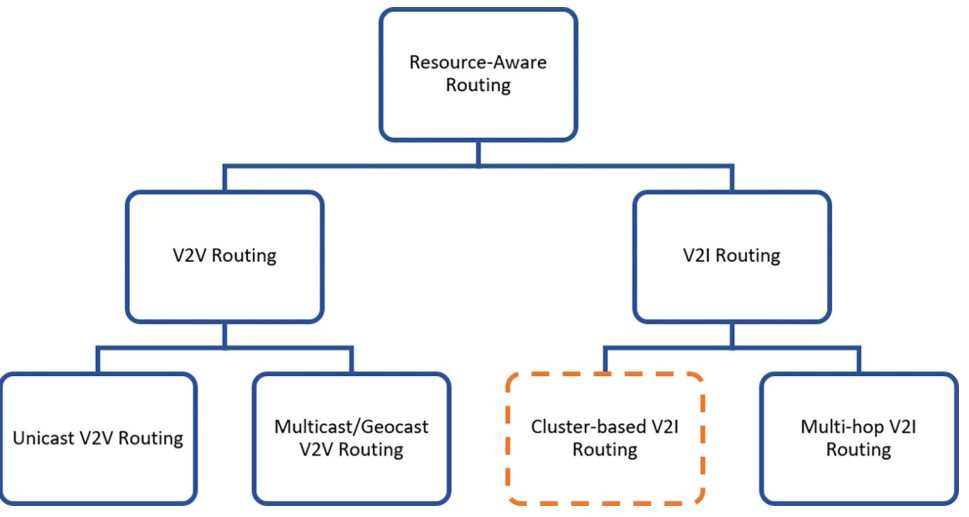

**Fig 1. Some possible scenarios for resource-aware routing in C-V2X.**

relayed by some vehicles along a path [12]. In these scenarios, resource-awareness is also essential as it may help better managing the clusters or multi-hop paths to improve the performance and efficiency of the C-V2X network. Fig 1 shows some possible scenarios where resource-aware routing seems promising.

In this paper, we focus on resource-aware routing in cluster-based V2I scenarios as highlighted in Fig 1. Although many resource management methods are proposed for V2I communications in C-V2X, it still continues to face the challenge of limited radio resources in some cells due to the unbalanced load of serving BSs. This challenge is more elaborated in 5G as we have ultra-dense networks and small cells there. This is while routing algorithms can help in resource management if those redirect the load of saturated cells to neighboring ones via modification of clusters. The reason why we do not alter the radio resources assigned to the cells (BSs) rather than changing the load of their clusters is that changing the resources assigned to a cell is not always suitable regarding the required QoS of users and reuse of resources in other neighboring cells. The reuse of radio resources in cellular networks imposes restrictions on the change of frequency bands assigned to cells. For example, as shown in Fig 2 (A), in a typical cellular network, each subchannel is reused in different cells as depicted by certain colors. Furthermore, the figure shows a loaded BS ($BS_1$ in the middle) which is loaded by three clusters. Moreover, there is no chance for $BS_1$ to borrow radio resources (RBs) from the neighboring BSs since some of them ($RB_5$, $RB_6$, $RB_7$) are loaded and the resources of others ($RB_2$, $RB_3$, $RB_4$) cannot be loaned as those are reused in nearby BSs and their use violates the interference limits. Consequently, we propose resource-aware cluster-based routing to solve the problem by re-clustering some clusters to balance the load as demonstrated in Fig 2(B). In next section, we will introduce our re-clustering method to solve this problem.

## Proposed method

As noted in the previous section, resource-aware cluster-based routing is an issue that has not been addressed in hybrid C-V2X/DSRC networks earlier. In this section, we propose a novel resource-aware cluster-based routing method for heterogeneous vehicular networks that employ both C-V2X and DSRC links. In the subsequent subsections, we will model and describe the aforementioned problem scenario and provide details of the proposed solution.

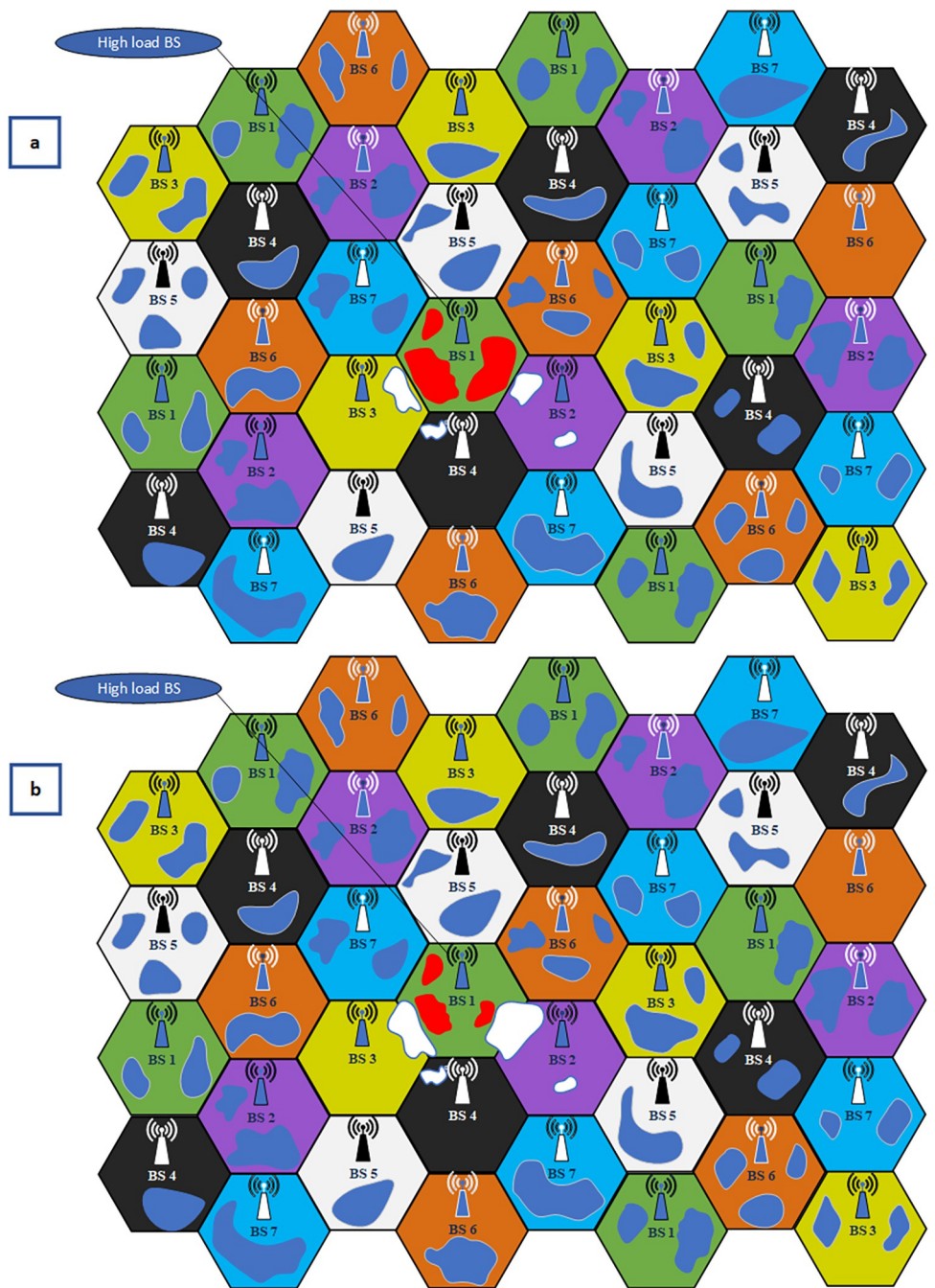

**Fig 2.** A cellular network with frequency reuse where clusters are presented as free shapes: (a) BS 1 (green area) loaded by three red clusters, (b) re-clustering is done and some of vehicles of red clusters are now served by white clusters associated to the lowest load neighbors, i.e., BS 2 and BS 3.

## System model

A vehicular communication area is assumed that contains some vehicles, denoted by $\{X_1, X_2, . . ., X_V\}$. We assume some (Macro or Small) BSs deployed at various points around the roads and crossroads, as denoted by $\{BS_1, BS_2, BS_3, . . ., BS_k\}$. Each vehicle can connect to a BS via direct link (using C-V2X resources) or indirect link (through a CH using DSRC and then

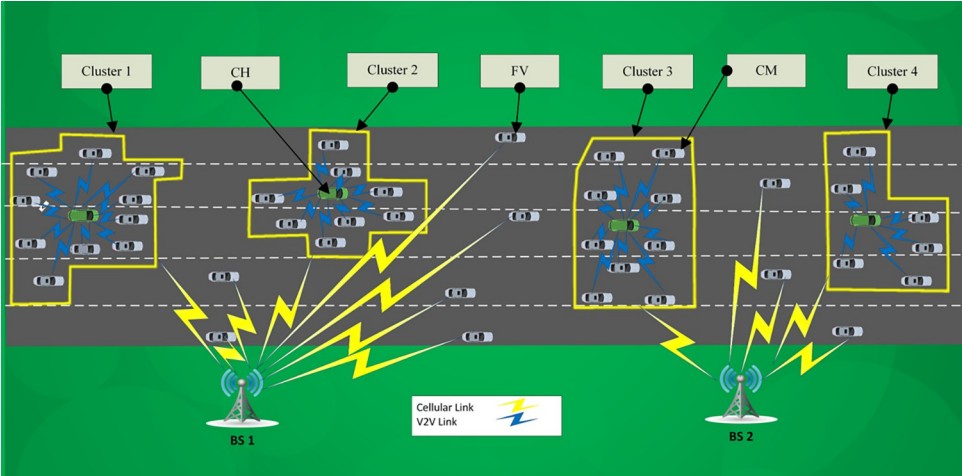

**Fig 3. Sample vehicular scenario after applying algorithm 1 ($\varepsilon$ = 50, MinPts = 6).**

C-V2X resources). We also assume a reuse-factor for the cellular network where the resources of a BS are reused in BSs with an acceptable distance from that BS. Fig 3 shows a sample scenario and Table 3 provides the notations considered in this paper. DBSCAN is used to construct clusters for one-hop routing because, in contrast to the mean-shift algorithm [25, 38], scattered vehicles outside the vehicle density range are considered. Also, it is not necessary to predefine the number of clusters, and it constructs clusters with unusual shapes in low time complexity [18]. The exploited DBSCAN-based clustering method groups vehicles into clusters $\{C_1, C_2, C_3,\ldots,C_h\}$ with high similarities in location. Each cluster has a cluster head from set $\{CH_1, CH_2, CH_3,\ldots,CH_h\}$, where $CH_1 \in C_1$, $CH_2 \in C_2$, and so on. FVs also connect to the BS directly using C-V2X links. A CH functions as the Relay Node (RN) for the CMs and redirects their traffic to a BS via C-V2X links. Each CH is associated with a BS (the one with highest Received Signal Strength (RSS)) and exploits a number of RBs to relay the packets of its CMs to the BS.

Using DBSCAN, vehicles are evaluated one by one concerning their surrounding vehicles to form clusters. The vehicles with sufficient neighboring vehicles around them, i.e., at least

**Table 3. Notations and their description.**

| Notation | Description |
|---|---|
| $\{X_1, X_2,\ldots, X_V\}$ | Set of vehicles in the system |
| $\{BS_1, BS_2\ldots, BS_K\}$ | Set of Base Stations assumed at the working area |
| $\{C_1, C_2\ldots, C_h\}$ | Set of clusters of the vehicles |
| $\{CH_1, CH_2\ldots, CH_h\}$ | Set of cluster heads where $CH_h \in C_h$ |
| $N_{min}$ | Minimum number of vehicles to construct a cluster using DBSCAN |
| $\varepsilon$ | Distance threshold of DBSCAN algorithm |
| $Max_{DSRC}^{distance}$ | Maximum communication range for DSRC links |
| $BS_{minload}$ | The BS with the lowest load adjacent to $BS_{maxload}$ |
| $BS_{maxload}$ | The BS with the highest load in the region |
| $\omega$ | Minimum acceptable Value for $N_{min}$ |
| $\gamma$ | The load difference percentage of two adjacent BSs |
| $load^{BS}_i$ | Load of $i^{th}$ BS |

$N_{min}$ vehicles within the radius of $\varepsilon$, are considered as core vehicles that are allowed to make clusters. Neighboring vehicles of vehicle $x$ are defined as [35]:

$$N_\varepsilon(x) = \{y \in D_{unprocessed} | distance(x, y) \leq \varepsilon\} \tag{1}$$

with size, $|N_\varepsilon(x)|$. Three types of vehicles are defined in the DBSCAN algorithm:

1. **Core vehicles**: vehicles like $x$ that support $|N_\varepsilon(x)| \geq N_{min}$.

2. **Border vehicles:** vehicles like $x$ that support $|N_\varepsilon(x)| < N_{min}$ and at least one of their neighbors is a core vehicle.

3. **Noise vehicles**: vehicles like $x$ that support $|N_\varepsilon(x)| < N_{min}$ *and none of their neighbors is a core vehicle.*

Fig 3 illustrates a sample DBSCAN-based clustering of vehicles consisting of CHs, CMs, and FVs. We notice that the CMs are connected to CHs using DSRC and cellular resources are used for V2I connections of CHs and FVs. The FVs are scattered vehicles that are tagged as noise in DBSCAN. The clusters in Fig 3 are highlighted using polygons. The blue communication link is used for intra-cluster communications whereas the yellow communication link is used for V2I connections.

The DBSCAN clustering that is taken into consideration by our system is displayed in Algorithm 1. When using this approach of clustering, the choice of clusters is made in such a way that every CH covers all of its members with respect to DSRC coverage. Even though this clustering strategy improves the efficiency by more effectively reusing the cellular bands, the problem comes when the BS that serves a CH has a limited number of resources to support the traffic of all the CMs associated to that CH. Additionally, as explained in Introduction section, we assume that the aforementioned BS is unable to simply borrow additional resources or alter the spectrum reuse due to the interference that will be imposed to other parties.

```
Algorithm 1. DBSCAN-based vehicle clustering
1. Input: N_min, ε, Vehicles
2. Output: clusters
3. D_unprocessed ← Vehicles
4. Obtain N_min from Eq (2)
5. ε = ⟦Max⟧_DSRC^distance
6. no_of_clusters = 0
7. While (D_unprocessed ≠ ∅) Do
8.    For each vehicle x ∈ D_unprocessed
9.    IF |N_ε(x)|<N_min and ∄ y∈{N_ε(x)} that y is a core vehicle Then
10.        Mark x as noise vehicle
11.        no_of_noise_vehicles ++
12.        D_unprocessed ← D_unprocessed-{x}
13.      Else IF |N_ε(x)|<N_min and ∃ y∈{N_ε(x)} that y is a core vehicle
14.          Mark x as a border vehicle
15.      Else IF |N_ε(x)|≥N_min and ∄ y∈{N_ε(x)} that y is a core vehicle
16.          Mark x as a core vehicle
17.          no_ of_ clusters ++
18.          D_DR(x) ← all distance_reachable vehicles from x that are in
D_unprocessed
19.          Clusters_no_of_clusters←{x}+D_DR(x)
20.          D_unprocessed←D_unprocessed-Cluster_no_of_clusters
21.      End IF
22.    End For
   End While
```

## Proposed re-clustering algorithm

This section discusses a resource-aware re-clustering algorithm that adaptively alters the settings of the DBSCAN clustering mechanism regarding the radio resource availability of BSs. The initial value for the parameter $\varepsilon$ in Algorithm 1 is decided depending on the maximum coverage that can be established for DSRC communications, i.e., $Max_{DSRC}^{distance}$. For initializing $N_{min}$, we consider non-overlapping circular regions with a radius $Max_{DSRC}^{distance}$ in the map, and select the area with a minimum number of vehicles (but higher than or equal to a constant number, $\omega$) to adjust the initial value. Eq (2) indicates an initial value of $N_{min}$ assuming $l$ circular regions in the environment,

$$N_{min}^{initial} = \max(\min(S_1, S_2, \ldots, S_l), \omega) \tag{2}$$

where $S_l$ is the number of vehicles in $l^{\text{th}}$ region.

Clusters of vehicles are susceptible to QoS degradations because of the uncontrolled size of those clusters regarding the available resources of serving BSs. To address this issue, we will present a resource-aware DBSCAN-based re-clustering strategy. As indicated in Algorithm 2, the first thing that we do is searching the network for BSs that have a high load because these are likely to be locations of inconsistency. Using the equations below, we can determine that there are two clusters, one belonging to the low load BS and the other belonging to the high load BS, for any high load BS that is near to a low load BS

$$C_{minload} = argmin_{c \in C | CH_c \bowtie BS_{minload}}(d_{CH_c, BS_{maxload}}) \tag{3}$$

$$C_{maxload} = argmin_{c \in C | CH_c \bowtie BS_{maxload}}(d_{CH_c, BS_{minload}}) \tag{4}$$

where $C_{minload}$ and $C_{maxload}$ refer to the selected clusters associated with low load BS ($BS_{minload}$) and high load BS ($BS_{maxload}$), respectively. As seen, these clusters are selected based on their distances ($d$) to the other BS than the one they have associated with (association is denoted by $\bowtie$).

Algorithm 3 demonstrates the proposed DBSCAN-based re-clustering algorithm, known as DBSCAN 2, which is intended to reduce the significant load disparity between nearby BSs concerning the availability of cellular resources. Consequently, the DBSCAN parameters that can be configured, i.e., $\varepsilon$ and $N_{min}$, are re-calculated depending on the resources that have been assigned to each BS to accommodate the most equitable use of the resources that are available to the BSs. The following equations are proposed to determine the parameters that are used for re-clustering. These parameters are calculated independently for each pair of minimum load and maximum load BSs. $N_{min}^{BS_{minload}}$ and $N_{min}^{BS_{maxload}}$ are obtained from Eqs (5) and (7), for nominated clusters associated with $BS_{minload}$ and $BS_{maxload}$, respectively. In these equations, $|C_{minload}|$ and $|C_{maxload}|$ are the size of clusters obtained from Eqs (3) and (4) and are associated with $BS_{minload}$ and $BS_{maxload}$, respectively. Furthermore, new $\varepsilon$ values, i.e., $\varepsilon^{BS_{minload}}$ and $\varepsilon^{BS_{maxload}}$, are obtained for those clusters from Eqs (6) and (8), respectively. The two parameters obtained from Eqs (5) and (6) are used for minimum load BS to increase its clusters' size, while the ones obtained from Eqs (7) and (8) are used for maximum load BS to decrease its clusters' size. Therefore, some of the vehicles associated to $BS_{maxload}$ through their CH will be connected to the minimum load BS ($BS_{minload}$) after re-clustering. Therefore, we have more balanced clusters.

$$N_{min}^{BS_{minload}} = \left(\frac{load_{BS_{maxload}} - load_{BS_{minload}}}{load_{BS_{maxload}}}\right) * (|C_{maxload}| - |C_{minload}|) + N_{min}^{initial} \tag{5}$$

$$\varepsilon^{BS_{minload}} = \frac{1}{2}\max(\Delta d_{W,Z}(C_{minload})) + \left(\left(\frac{load_{BS_{maxload}} - load_{BS_{minload}}}{load_{BS_{maxload}}}\right)*\varepsilon^{initial}\right) \tag{6}$$

$$N_{min}^{BS_{maxload}} = N_{min}^{BS_{minload}}*\left(\frac{load_{BS_{maxload}} - load_{BS_{minload}}}{load_{BS_{maxload}}}\right) \tag{7}$$

$$\varepsilon^{BS_{maxload}} = \varepsilon^{BS_{minload}} - \left(\left(\frac{load_{BS_{maxload}} - load_{BS_{minload}}}{load_{BS_{maxload}}}\right)*\varepsilon^{BS_{minload}}\right) \tag{8}$$

**Algorithm 2.** Finding load inconsistency locations in clustered C-V2X/
DSRC network
**1. Input:** set of BSs {$BS_k$}, the number of BSs ($n$), overload threshold
($\theta$), BSs' loads {$load^{BS}_k$}, set of current clusters ($C$)
**2. Output:** updated set of clusters ($C$)
**3.**  For *i = 1 to n* **Do**
**4.**    **IF** $load^{BS}_i$>$\theta$ **Then**
**5.**     **For *j = 1 to n* Do**
**6.**       **IF** $i{\neq}j$ **Then**
**7.**        **IF $BS_j$ is adjacent to $BS_i$ & ($load^{BS}_i$–$load^{BS}_j$)$\geq\gamma$ Then**
**8.**               $C_{maxload} \leftarrow argmin_{c\in C|CH_c\rhd\lhd BS_i}(d_{CH_c,BS_j})$
**9.**               $C_{minload} \leftarrow argmin_{c\in C|CH_c\rhd\lhd BS_j}(d_{CH_c,BS_i})$
**10.**               $C = $**Call**D BSCAN2$(i, j, C_{maxload}, C_{minload})$
**11.**          **End IF**
**12.**        **End IF**
**13.**      **End For**
**14.**    **End IF**
**15.**  **End For**
**16. End**
**Algorithm 3.** Resource-aware DBSCAN-based re-clustering algorithm
(DBSCAN 2)
**1. Input:** $BS_{minload}$, $BS_{maxload}$, $C_{minload}$, $C_{maxload}$
**2. Output:** Clusters ($C$)
**3.** $C = \{C_{minload}, C_{maxload}\}$
**4.** $D_{unprocessed}\leftarrow\{Vehicles\ of\ C_{minload}\}$
**5.** $D_{unprocessed}\leftarrow D_{unprocessed}+\{Vehicles\ of\ C_{maxload}\}$
**6.** $N_{min}^1 = N_{min}^{BS_{minload}}$ based on Eq (5)
**7.** $N_{min}^2 = N_{min}^{BS_{maxload}}$ based on Eq (7)
**8.** $\varepsilon^1 = \varepsilon^{BS_{minload}}$ based on Eq (6)
**9.** $\varepsilon^2 = \varepsilon^{BS_{maxload}}$ based on Eq (8)
**10.** *no_of_clusters* = 0
**11. While** ($D_{unprocessed} \neq \emptyset$) **Do**
**12.**  **For** vehicle x $\in$ D$_{unprocessed}$ **Do**
**13.**    **IF** $distance[x, BS_{minload}]) < BS_{minload}^{coverage}$ **Then**
**14.**       $N_{min} = N_{min}^1$
**15.**       $\varepsilon = \varepsilon^1$
**16.**  **Else**
**17.**       $N_{min} = N_{min}^2$
**18.**       $\varepsilon = \varepsilon^2$
**19.**    **End IF**
**20.**    **IF** $|N_\varepsilon(x)|$<$N_{min}$ and $\nexists$ y$\in\{N_\varepsilon(x)\}$ that *y* is a core vehicle **Then**
**21.**      Mark *x* as ***noise vehicle***
**22.**      *no_of_noise_vehicles ++*

```
23.        D_unprocessed ← D_unprocessed−{x}
24.    Else IF |N_ε(x)|<N_min and ∃ y∈{N_ε(x)} that y is a core vehicle
Then
25.      Mark x as a border vehicle
26.    Else IF |N_ε(x)|≥N_min Then
27.      Mark x as a core vehicle
28.      no_of_clusters ++
29.      D_DR(x)←all distance reachable vehicles (borders) from x in
D_unprocessed
30.        Clusters_no_of_clusters←{x}+D_DR(x)
31.        D_unprocessed←D_unprocessed−Clusters_no_of_clusters
32.    End IF
33.  End For
34. End While
```

In above equations, $\Delta d_{W,Z}(C_{minload})$ is the distance between the two furthest vehicles ($W$ and $Z$) in the $C_{minload}$.

## Clarifying example

We proposed a method to manage the load of BSs by adjusting the clustering parameters and re-clustering the vehicles according to the resource availability of BSs. In contrast to previous works like [26], we do not alter radio resources assigned to BSs to manage the load inconsistency of BSs. This is due to the fact that radio resources assigned to each BS may be reused elsewhere by other BSs, and the change in resource mappings does not always have an optimum solution as justified in previous section.

To illustrate the proposed solution, we consider a scenario in which two BSs partially cover a junction (for simplicity). The area, as seen in Fig 4, comprises a route which is covered by $BS_1$ and $BS_2$. We presume that vehicles will always connect to the closest base station. Assuming that the heterogeneous network exploits Algorithm 1, two large clusters are constructed and linked to $BS_1$ via respective CHs as seen in Fig 4(A). However, $BS_1$ is overloaded in terms of the radio resources required by the two connected CHs, resulting in significant packet loss and high delay. Here, we assume that loaded BS, i.e., $BS_1$ has neighboring BSs that reuses the same channels as $BS_2$ and are loaded. So, if we transfer some radio channels from low-density BS, i.e., $BS_2$, to high density BS, $BS_1$, there is a high chance that the users in neighboring BSs experience high interference. It is while our proposed resource-aware re-clustering solution effectively alleviates the excessive load of $BS_1$ by modifying the clusters. Here, we apply DBSCAN 2 to reconstruct specific clusters connected to $BS_{maxload} = BS_1$ and $BS_{minload} = BS_2$. It is executed based on the available resources of BSs, with just the cluster sizes being altered. In this situation, $BS_1$ is deemed to be overloaded, whereas $BS_2$ is assumed to be functioning at half capacity (the difference in load between the two BSs is greater than presumed $\gamma$). Therefore, the clusters from $BS_1$ and $BS_2$ that are closest to their respective BSs are selected and re-clustering is executed. The new parameters of DBSCAN clustering are adjusted such that the size of the selected clusters from $BS_{minload}$ be increased while the size of the selected clusters from $BS_{maxload}$ be decreased. Fig 4(B) shows the new clusters after applying DBSCAN 2. In other words, if we pay attention to the cellular radio resources that are available in the cells, we will be able to solve the load inconsistencies to some extent by using re-clustering, and we will be able to do so without a significant change in the co-channel interference of the C-V2X network.

Fig 5 provides a concise walkthrough of the suggested method for viewing convenience. As seen, during initialization, the basic DBSCAN-based clustering is executed, but later, Algorithm 3 is executed after finding each two unbalanced BSs with candidate clusters (the clusters

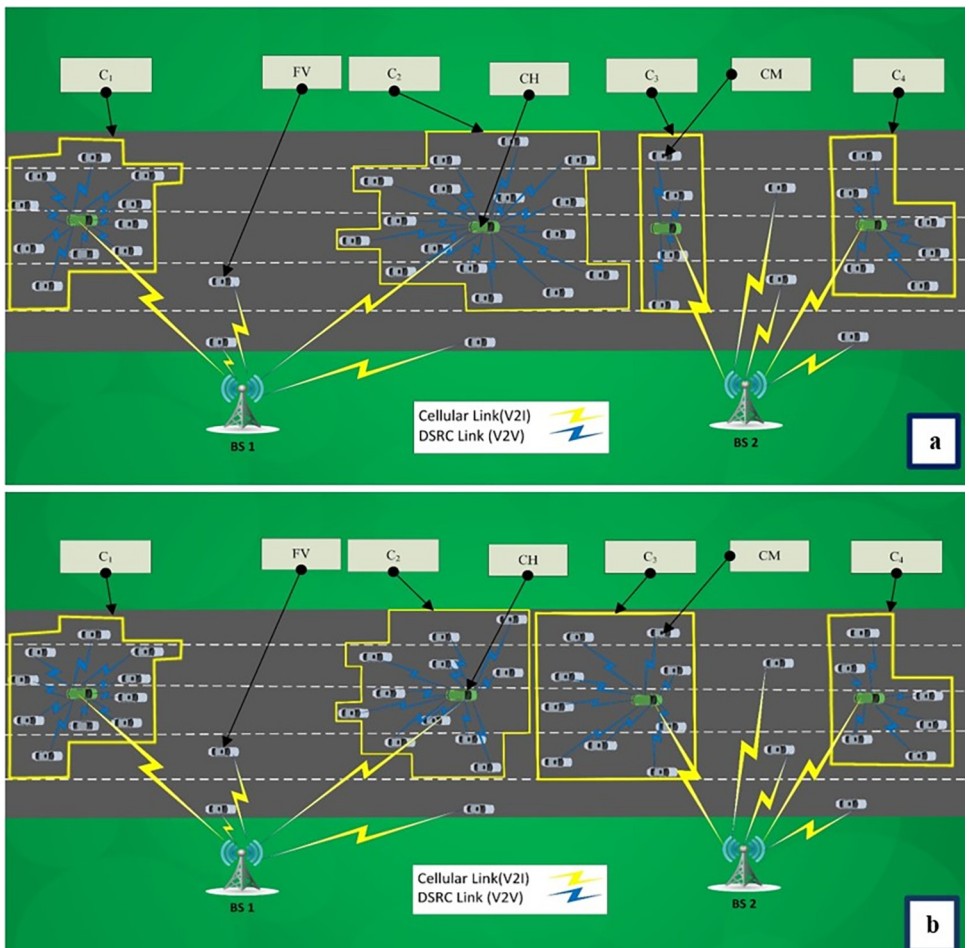

**Fig 4.** A part of VANET area with an inacceptable difference in BSs' loads, a) After DBSCAN, b) After DBSCAN 2.

obtained from lines 8 and 9 of Algorithm 2) to update the clusters in each round. For basic DBSCAN algorithm (Algorithm 1), the time complexity is of $O(V \log V)$ using the best implementations [12, 39] paying attention to the fact that $D_{unprocessed}$ includes all vehicles. However, each round of our re-clustering algorithm is of $O(|D_{unprocessed}| \times \log |D_{unprocessed}|)$ where $|D_{unprocessed}| = V$ in the worst case and so the overall time complexity of Algorithm 2 is of $O(nV \log V)$ in the worst case. Therefore, the complexity of proposed re-clustering is not considerably higher than basic DBSCAN as the number of BSs ($n$) is not significant compared to $V$. Our method leads to some handovers between neighboring BSs due to cluster changes in the first round where the clusters are re-arranged after initialization. But in next iterations, the initial DBSCAN-based clustering is not performed (as is obvious from Fig 4) and so the clusters are not changed significantly except due to vehicles' mobility and propagation of load inconsistency.

## Simulation results

Using a custom system level MATLAB simulator, the performance of the suggested technique is examined in this section. The simulated traffic conditions correlate to the datasets of the Next Generation Simulation (NGSIM (Program, which include a variety of genuine roadway

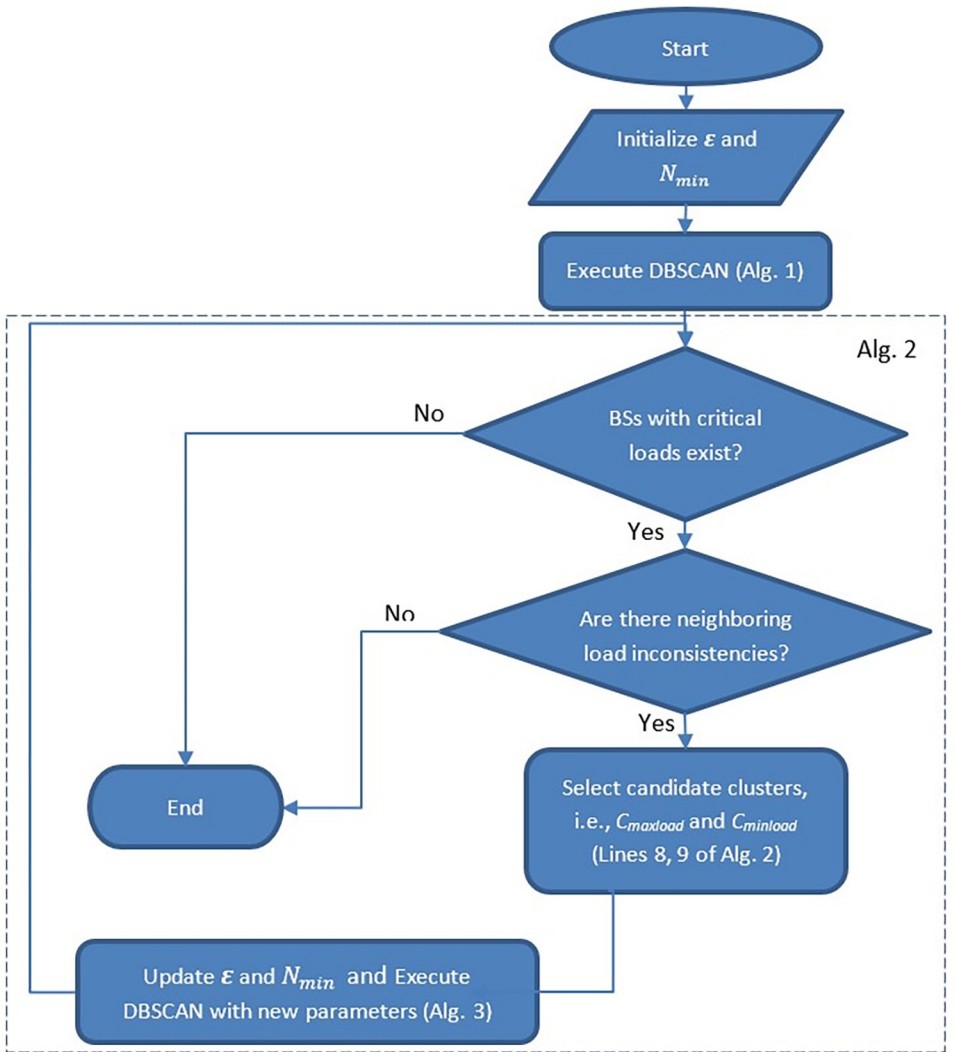

**Fig 5. The flowchart of the proposed method.**

settings [40]. We have selected diverse segments of I-80, US101, Peachtree, and Lanker Shim. These portions of the dataset have been scaled to $5000_m$, five-lane highways. Simulations are performed to examine the robustness of the suggested method utilizing various settings for the proposed DBSCAN-based algorithm in conjunction with various congestions (i.e., 1000, 2000 and 3000 vehicles). The communication range for DSRC is assumed 450 meters. Table 4 displays the simulation's parameters.

The strategy described in [37], is compared with the method that we have suggested in this article. The Mobility Management Entity (MME) and the Traffic Efficiency Server (TES) are reportedly responsible for managing resource allocation, as stated in [37]. The MME determines the position of the automobiles, whereas TES supplies BS with the appropriate resources. The BSs submit requests for resources to TES. TES assesses the number of resources necessary for each location and advises the micro-BSs to shift resources to the BSs that demand more. In our suggested methodology, there is no change to the allocation of resources; nevertheless, by re-clustering the clusters connected to BSs, we seek to manage excessive load

**Table 4. Simulation settings.**

| Parameter | Value |
|---|---|
| Number of eNodeB | 3 |
| Number of vehicles | 1000–3500 |
| $Max_{DSRC}^{distance}$ | 450 m |
| *MinPts* | 12–18 nodes |
| $\omega$ | 7 vehicles |
| $\theta$ | 90% of available RBs |
| $\gamma$ | 25% of available RBs |
| Data rate of each vehicle | One packet per each time step |
| Number of runs with different random seed | 250 |
| Number of available Resource Blocks | 300 |

conditions. Instead of transferring resources [37], our proposed technique modifies the size of the clusters. The evaluation metrics we used and the results of our comparisons are outlined below:

- *Spectrum efficiency*

  The reuse is used as a major indicator of spectrum efficiency, which is calculated from

$$reuse\ indicator(RI) = \frac{RA}{TR} \tag{9}$$

where *RI* represents the reuse indicator, *RA* represents the resource blocks allotted to cluster heads and independent cars for their vehicular traffic, and *TR* represents the total resource blocks accessible to vehicles. In Fig 6, we contrast the traditional method [37], with our suggested method in terms of *RI*. $sub_1$, $sub_2$, and $sub_3$ are identified as sub-resource pools 1, 2, and 3, respectively. As can be seen, our proposed method yields a higher *RI* and, consequently, a higher spectrum efficiency. It is because we attempt to restructure the clusters as opposed to changing the resource assignments, resulting in more efficient use of resource blocks for the delivery of packets from CHs to cellular infrastructure.

- *Packet Delivery Ratio (PDR)*

  The third measure to consider is the ratio at which packets are delivered, which may be calculated using Eq (10). Fig 7 depicts a comparison of the method of [37], to the way that we

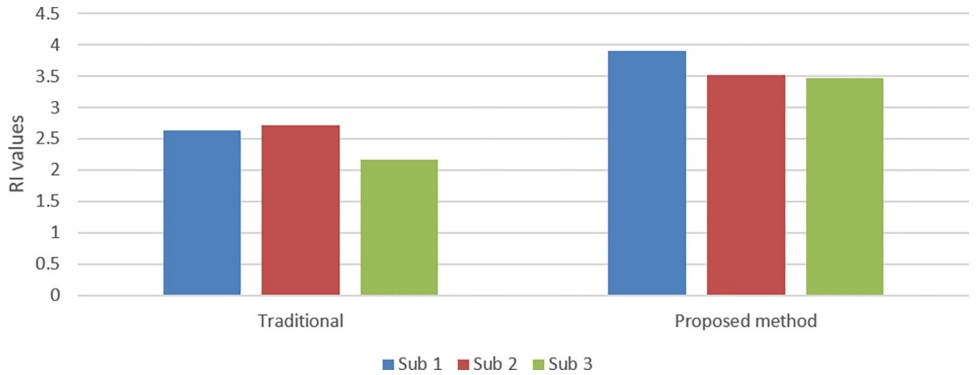

**Fig 6. Compression of traditional [37] and proposed method in terms of RI.**

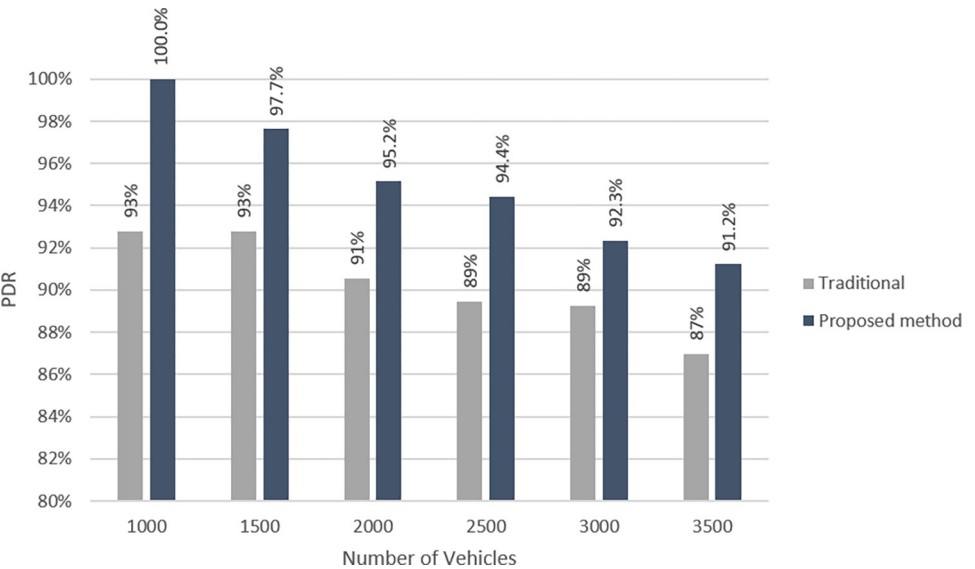

**Fig 7. PDR at different loads in traditional [37] and proposed methods.**

have proposed, in terms of *PDR*. This comparison is based on the assumption that each vehicle delivers one packet during each period and each packet is transmitted in one RB.

$$PDR = \frac{Received\ Packets}{Total\ Sent\ Packets} \tag{10}$$

We also compare the proposed method to the method of [37] in terms of the number of vehicles that their required cellular resources are supported by the cellular network. As shown in Fig 8, the proposed method serves more vehicles than previous method in all settings. It is due to the fact that it tries to balance the load of overloaded BSs by re-clustering their vehicles which leads to better support of their associated vehicles.

- *Load Difference (LDf)*

This statistic is derived from the load differential ratio that exists between each BS and the BSs that are located in its immediate vicinity. To determine the *LDf*, first, the load of each BS is determined by utilizing Eq (11), and then it is compared to the loads of the BSs that are its adjacent using Eq (12).

$$BSL = \frac{CL}{TAL} \tag{11}$$

$$LDf = \begin{cases} argmax_{BSL}\{BS_1, \dots, BS_k\} - argmin_{BSL}\{BS_1, \dots, BS_k\}, Max_{BSL} \geq \theta \\ 0, otherwise \end{cases} \tag{12}$$

where *CL* is the current load, *TAL* is the total acceptable load, *BSL* is the load of mentioned BS and the *LDf* is a measure of load balancing.

As depicted in Fig 9, assuming the resource management technique of [37], with the basic DBSCAN-based vehicle clustering, we discovered a significant amount of load variations between neighboring BSs ($BS_2$ and its neighbor BSs) due to the absence of resource availability consideration in the clustering solution. This is because, without analyzing the capacity of BSs in addition to clustering basics, we have an imbalance in the load of BSs, and changing the

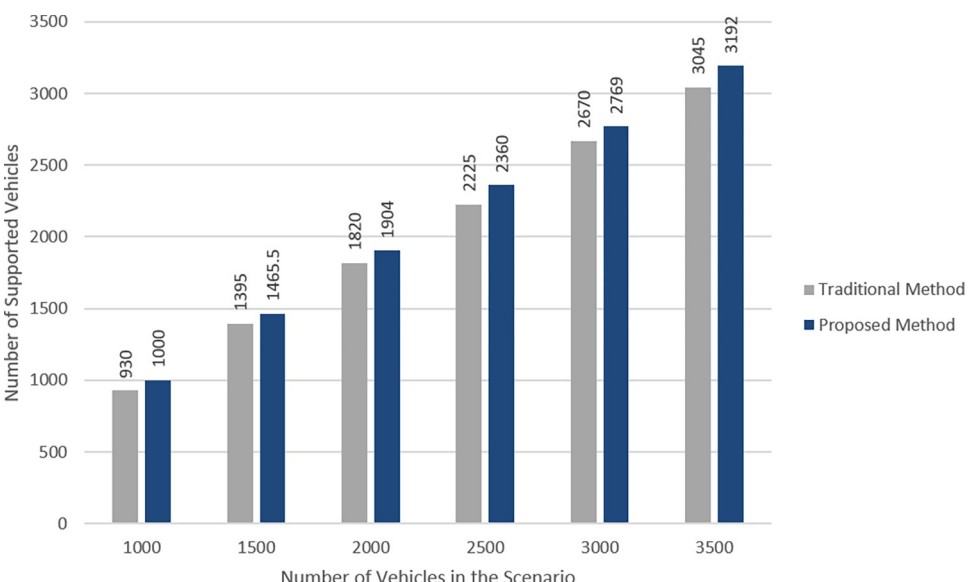

**Fig 8. Comparing the proposed method to the method of [37] in terms of the number of supported vehicles.**

assignment of resources to BSs may result in more inter-cell interferences, which in turn leads to a reduction in quality. In our suggested method, we take into account the size of clusters and the capacity of the BSs to balance the load of BSs through re-clustering to eliminate inter-cell interference and enhance performance.

The *LDf* between two nearby BSs (with the largest load difference) before and after re-clustering is depicted in Figs 10, and 11 respectively. In Fig 10, the right bar shows the initial load of the $BS_{minload}$ and the left bar shows the new load of $BS_{minload}$ after re-clustering. As seen, adjusting the load of BSs during re-clustering results in a more balanced load between two adjacent BSs. Here, $\rho$ indicates the maximum allowed load of each BS (a number of available RBs) where the $\theta$ is calculated as 90% of this load. In Fig 11, the left bar shows the initial load of $BS_{maxload}$ before re-clustering while the right bar shows the load value of $BS_{maxload}$ after applying the proposed method, i.e., transferring the extra load from $BS_{maxload}$ to $BS_{minload}$, which means that the two BSs' loads are more balanced.

Fig 12 compares the *LDf* of our suggested strategy to that of [37]. The left bars represent the *LDf* calculated using the approach described in [37], whereas the right bars represent the *LDf* calculated using the proposed method. The results indicate that the proposed strategy reduces the amount of *LDf*, and as a result, communications will be of greater quality.

## Conclusion

The Internet of Vehicles era presents some issues, one of which is the consideration of cooperative resource management and routing. In this paper, we proposed an enhanced DBSCAN-based re-clustering algorithm to better manage the resources of 5G cellular BS and mitigate load imbalance among BSs. The algorithm works by re-clustering the clusters that are located in between high load BSs and their adjacent low load BSs with a considerable load difference. The suggested mechanism modifies the settings of DBSCAN and re-clusters the associated vehicles to improve the clustering based on the resources that are made available by BSs. According to the findings, the improved DBSCAN-based technique functions more effectively than the existing method in terms of load balancing, PDR, spectrum efficiency, and support of

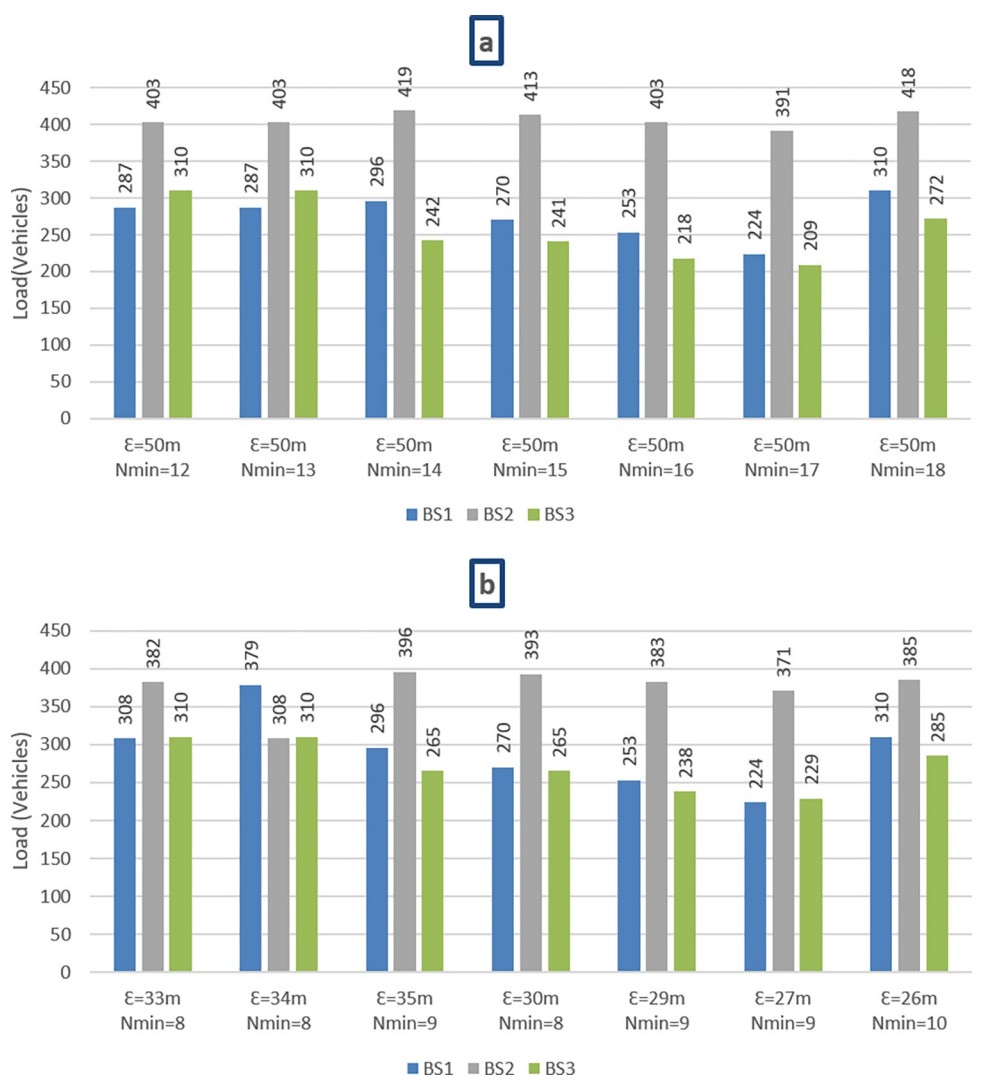

**Fig 9.** Adjacent BSs' loads at different $N_{min}$ and $\varepsilon$ values using traditional method [37] (a) and proposed method (b).

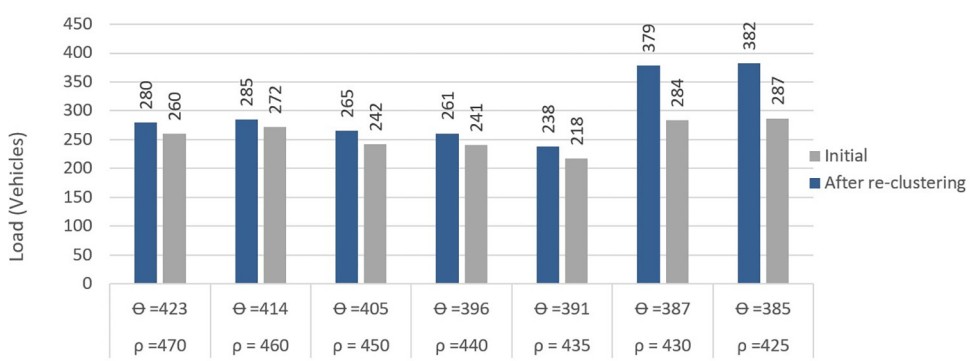

**Fig 10. Right bars are the initial $BS_{minload}$ values after DBSCAN while left bars are the values after using proposed method.**

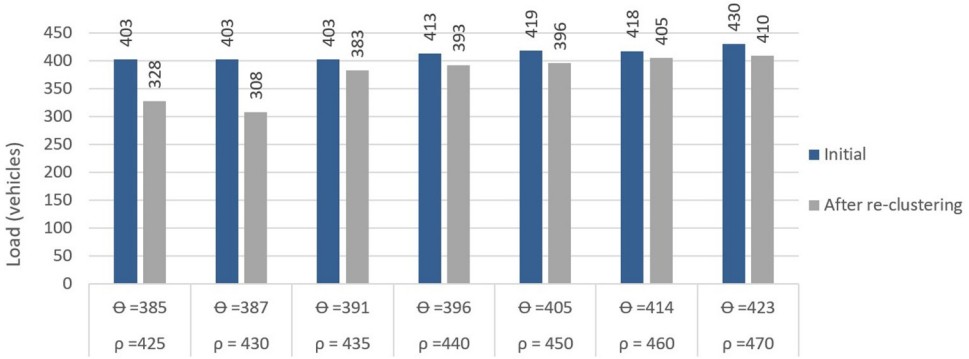

**Fig 11. Left bars are initial $BS_{maxload}$ after DBSCAN and right bars are new values using proposed method.**

the rate requirement of vehicles. However, the proposed method does not pay attention to the capacity of DSRC in supporting cluster members. Considering an analytical limit on the number of vehicles that DSRC can support in a cluster, and modifying the proposed method based on this limit, is addressed as a future work. Moreover, the lack of stability and overhead of maintaining the clusters are the shortcomings of the proposed method that should be improved in future work. Studying the impact of the proposed method on the number of handovers under various mobility patterns and reducing the handovers is a main future work. In addition, as a future work, the proposed resource-awareness concept can be investigated in

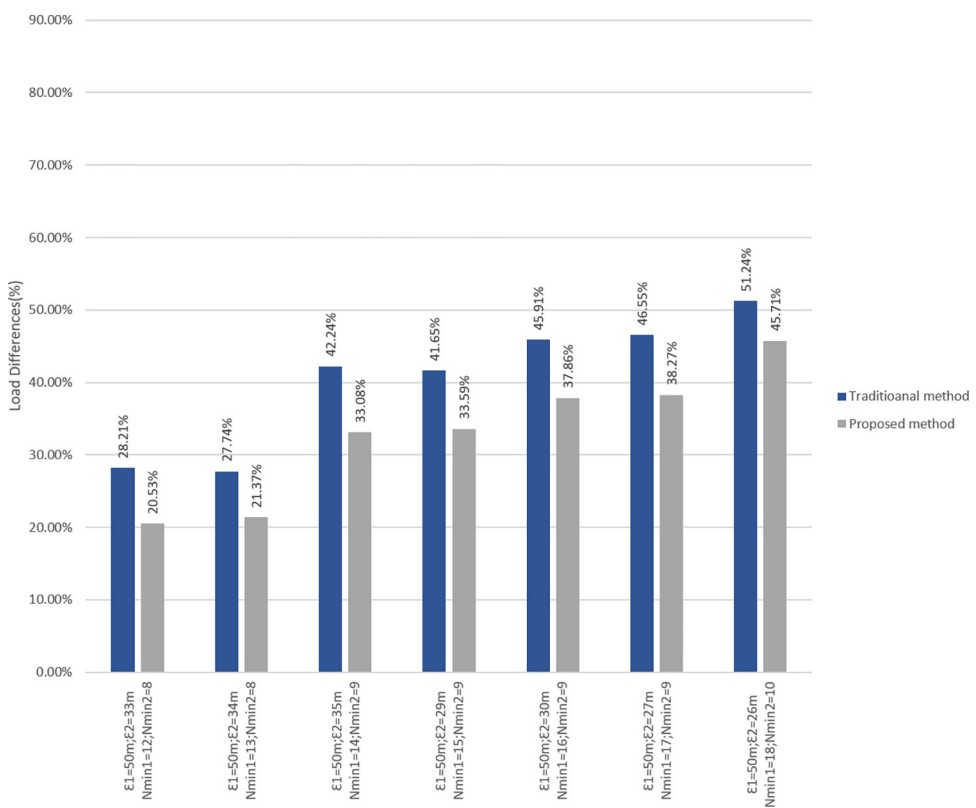

**Fig 12. left bars are LDf values of traditional method [37] while right bars are the LDf values using proposed method.**

other V2I routing mechanisms and clustering algorithms. Also, we are planning to employ the proposed resource-aware re-clustering algorithm besides the dynamic geo-based resource management mechanism for V2V communications, as part of our future work.

## Author Contributions

**Conceptualization:** Behrouz Shahgholi Ghahfarokhi.

**Data curation:** Jaafar Sadiq Alrubaye.

**Formal analysis:** Jaafar Sadiq Alrubaye, Behrouz Shahgholi Ghahfarokhi.

**Investigation:** Jaafar Sadiq Alrubaye, Behrouz Shahgholi Ghahfarokhi.

**Methodology:** Jaafar Sadiq Alrubaye, Behrouz Shahgholi Ghahfarokhi.

**Project administration:** Behrouz Shahgholi Ghahfarokhi.

**Software:** Jaafar Sadiq Alrubaye.

**Supervision:** Behrouz Shahgholi Ghahfarokhi.

**Validation:** Jaafar Sadiq Alrubaye, Behrouz Shahgholi Ghahfarokhi.

**Visualization:** Jaafar Sadiq Alrubaye.

**Writing – original draft:** Jaafar Sadiq Alrubaye.

**Writing – review & editing:** Jaafar Sadiq Alrubaye, Behrouz Shahgholi Ghahfarokhi.

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
