## [Decision Letter · Decision Letter 0]

29 Aug 2023

PONE-D-23-25645Resource-Aware Cluster-based Routing in Hybrid C-V2X/DSRC Vehicular NetworksPLOS ONE

Dear Dr. Shahgholi Ghahfarokhi,

Thank you for submitting your manuscript to PLOS ONE. After careful consideration, we feel that it has merit but does not fully meet PLOS ONE’s publication criteria as it currently stands. Therefore, we invite you to submit a revised version of the manuscript that addresses the points raised during the review process.

We look forward to receiving your revised manuscript.

Kind regards,

C. Suganthi Evangeline, B.E.,M.E.,Ph.D

Academic Editor

PLOS ONE

Journal Requirements:

4. Please ensure that you refer to Figure 2 in your text as, if accepted, production will need this reference to link the reader to the figure.

5. Please include a copy of Table 2 which you refer to in your text on page 15.

Additional Editor Comments:

The manuscript needs to be revised before I can recommend publication in this journal. The quality of the work needs to be improved and comments have been sent to the authors in this regard. I have collected four qualified reviews by now. The reviewers raised serious concerns on the novelty and the contribution of this work. First, the related works, including the state-of-the-art clustering algorithms, are not discussed fully, making the contribution of this work not easy to define; second, the comparison between the proposed method and existing ones is insufficient, which is also criticized by the reviewer(s).

Reviewers' comments:

Reviewer's Responses to Questions

**Comments to the Author**

1. Is the manuscript technically sound, and do the data support the conclusions?

Reviewer #1: Yes

Reviewer #2: Partly

Reviewer #3: Yes

Reviewer #4: Partly

2. Has the statistical analysis been performed appropriately and rigorously? 

Reviewer #1: Yes

Reviewer #2: No

Reviewer #3: Yes

Reviewer #4: Yes

3. Have the authors made all data underlying the findings in their manuscript fully available?

Reviewer #1: Yes

Reviewer #2: No

Reviewer #3: Yes

Reviewer #4: Yes

4. Is the manuscript presented in an intelligible fashion and written in standard English?

Reviewer #1: Yes

Reviewer #2: No

Reviewer #3: Yes

Reviewer #4: No

5. Review Comments to the Author

Reviewer #1: As a reviewer, I read this article several times and found the topic very interesting and timely. Further, I found that this paper is well-written and organized and deals with a very interesting approach. However, below are some of the comments that I think the authors need to address before accepting the paper. The aim of this comment is to improve the quality of the paper.

• The title is informative and relevant, but it sounds not interesting. I suggest rewriting the title to be more interesting.

• The aim of the research is clear and the main article matches the abstract. However, it is not very clear what the study found. Therefore, I suggest writing the research result.

● I suggest writing the main contribution of the paper in the introduction section.

● It would be better to make a comparison table for related work and explain how the proposed research work differs from the existing one.

● The data results in Figure 5, Figure 6, and Figure 7 are relevant and presented clearly. However, it would be better to provide more analysis and discuss the results from multiple angles.

• All related and recent references should be seen. For example, the following reference is closely related and I recommend citing them:

http://ijece.iaescore.com/index.php/IJECE/article/view/24016

https://online-journals.org/index.php/i-jim/article/view/26663

● The conclusion answers the aims of the study and is supported by results, but it is missing future work. Therefore, it would be better to inform future research opportunities.

Reviewer #2: In this paper, the author proposes Resource aware Cluster-based Routing for C-V2X/DSRC Vehicular communication. The reviewer has several concerns as follows:

1. All abbreviations should be defined in the first place, eg., IEEE, 4G, LTE, 3GPP, UE, RSU, C-V2X..and so on.. vehicle-to-vehicle should be V2V defined in Introduction. In many places abbreviation and acronyms are missed out. The same acronym in abstract and another part of the paper should be defined in abstract and another place in the first place.

2. What are the main challenges in resource-aware cluster based routing for vehicular networks? In order to appeal to interested readers, the authors need to point out the underlying difficulties in the studied work.

3. The study found that few existing algorithms consider negative effects induced by resource management in cellular networks for c-v2x communication. How to evaluate the rationality of the designed criteria considering the kinds of factors of ineffective HO in this study?

4. In this work how to ensure that the quality requirements are satisfied?

5. To provide enough and more information about various communication technologies and brief background about VANET, the following articles are suggested to be used and cited in place of ref 1, 2 in the Introduction.

New References

[1] A two-phase fuzzy based access network selection scheme for vehicular ad hoc networks. Peer-to-Peer Networking and Applications. (https://link.springer.com/article/10.1007/s12083-021-01228-w)

[2] Two-phase access network selection scheme based on weighted sum and game theoretical approaches for vehicular ad hoc networks. Journal of Circuits, Systems and Computers, 30(11), 2150206. (https://www.worldscientific.com/doi/abs/10.1142/S0218126621502066)

The following article is suggested to be cited instead of reference [11]

[11]. An efficient data transmission in VANET using clustering method. International Journal of Electronics and Telecommunications. (https://journals.pan.pl/dlibra/show-content?id=106023)

6. Compared with the existing work, what are the key research gaps and contributions of this paper? The authors are suggested to clarify their contribution's specific research problem and novelty in the Introduction Section. Moreover, the reviewed related works in Section II should correspond to the research gaps. Explore the related works in more detail with 3 dimensions: Resource aware, Clustering and Routing. The related work should be re-analysed and reorganised. In related works under the theme routing, I suggest to include the below article instead of Ref. [15]

[15] A Fuzzy Based Trust And Priority Enabled AODV Routing Scheme For Vehicular Network. INTERNATIONAL JOURNAL OF SCIENTIFIC & TECHNOLOGY RESEARCH.

7. The proposed System model is vague for the reader. Including an illustrative figure for better understanding of the system scenario is highly suggested.

8. The proposed model has 3 variants (3 Algorithms). Why are they necessary? How do they stand during experimental comparison/evaluation?

9. How does the proposed resource-aware cluster based routing approach compare to existing methods in terms of performance and complexity?

10. The authors have well introduced the proposed work, however, the downside of this proposed work is not mentioned in detail. Authors should give a brief explanation related to the disadvantages of the proposed method, and also the main challenges (other than the computational complexity).

11. What do you mean by middle in [16] in Introduction section.

12. Why do you go for DBSCAN clustering mechanism when there are several other mechanisms?

13. Proper references should be provided for the equations that were borrowed from the literature.

14. There are lot of figures without much necessity or explanation in the paper, e.g., Figure 1, 2 which makes the paper less readable. Remove the above figures. The authors are suggested to draw an illustrative figure about clustering in VANET scenario in Introduction section.

15. The comparison between proposed method and traditional method in Figure 5, 6 is insufficient. What is the “traditional method” that is mentioned of? Make comparisons with existing other approaches to validate the results.

16. There remain many typos and grammatical errors. The writing of the paper needs to be improved from informal to formal way. (eg., A question may arise…), please reframe such sentences. Hence it is requested to proofread your text.

Reviewer #3: Comments:

1. It's interesting to note that DBSCAN, being a well-established clustering algorithm, has been part of the field for a while. The inclusion of re-clustering to form a new methodology adds a fresh perspective to its application. Considering your innovative approach, I wonder if there's room for exploring and producing new clustering concepts within this methodology. This could potentially push the boundaries even further and contribute to the advancement of the field such as grid based , ordering points based ,hierarchy based etc…… It will improve the Quality of ur paper.

2. The article quite informative and appreciate the coverage of the contents. However, it seems that there's a gap in the expansion of certain concepts like DRSC and LTE. Including these would provide a more comprehensive view and enhance the reader's (new) understanding.

3. The references and citations predominantly span up until the year 2015 (avoid 2007). To provide a more up-to-date context and incorporate recent advancements, I recommend considering references beyond that year. This would enhance the relevance and currentness of the article.

4. This article addresses the challenge of unbalanced load. Could you kindly provide more insight into how this is achieved through enhanced DBSCAN approach? Specifically, I'm interested in understanding the separate methodology you've developed to balance the load within networks. Elaborating on this aspect would greatly enhance the practical value of the research.

5. the manuscript shows promise in addressing load balancing, clustering, and re-clustering. To enhance clarity, consider adopting a module-wise approach. Clearly outline each module's purpose, methodologies with equations , benefits, and practical cases with suitable diagrams. This will provide readers a systematic understanding. Looking forward to the refined manuscript.

6. The manuscript has highlighted important concerns associated with frequent re-clustering, such as resource intensity, over-fitting, maintenance challenges, and lack of stability. Addressing these challenges is essential for the practical application of frequent re-clustering. Could you elaborate on potential strategies or methodologies that can be employed to overcome or mitigate these drawbacks?

7. In this article, eNodeB is not discussed except in the table. Could you provide a brief explanation of what eNodeB refers to and its importance?

Reviewer #4: The paper seems to be very timely and focuses on a topic that is very paramount as it relates to Vehicle to Everything (V2X) communication in 5G wireless networks. In V2X communications, the issue of resource availability, latency and stability have posed several challenges especially when it comes to IoVs. The topic presented is good, but this paper still suffers from major issues as given below:

1. The authors should improve on the literature review section of this manuscript. In section II, a table should be developed which compares the contributions of this work to other recent manuscripts in this field especially papers where 5G wireless networks have been used with IoVs technologies for wireless signal propagation or a similar scenario. The focus and coverage of the work, its limitations should also be included in the table.

2. The abstract needs to be re-written and some sentences needs to be rephrased. For example, this sentence needs to be rephrased “It is while the routing protocols could help redirect the load of busy BSs to neighboring BSs if those protocols were resource-aware”

”Traditional cluster-based routings do not attend the resource availability of cellular BSs in their decisions”

Please rephrase those sentences and many more so they make meaning.

3. There are many grammatical errors and incomplete sentences in the manuscript. The authors should correct those uncompleted sentences

4. In figure 1, Is this figure original or copied from somewhere? If copied, appropriate citation should be added. Same holds for Figure 2.

5. The contribution of this work needs to be well spelt out in the manuscript. List the contributions of this work at the end of the introduction in Section I.

6. The methodology section of this work need to be properly enhanced. In section III, the proposed model needs to be well developed. The authors should expand on this section and show all the stages of the model development. All variables for the model equations should be well defined and explained as it relates to this study.

7. Nowadays, the proliferation of massive wireless devices enables the transmission of sensitive user information over open communication channels. However, the security and privacy of such critical data have been worrisome. How do you guarantee the protection of sensitive data in IoVs communications? One would expect that this important aspect should be given significant consideration in the current work. What do you think?

8. In the conclusion section, a brief summary of the key findings from the work is requested. It is imperative to state the key takeaways from the work and their implications

9. List of abbreviations table should be included in this work. Some terms have been used without the authors defining those terms.

10. More authoritative references should be cited. The references need to be increased.

6. PLOS authors have the option to publish the peer review history of their article (what does this mean?). If published, this will include your full peer review and any attached files.

Reviewer #1: **Yes: **Ass. Prof. Ali H. Wheeb

Reviewer #2: **Yes: **Ashmiya Lenin

Reviewer #3: **Yes: **Dr.J.Naskath, Associate Professor,National Engineering College ,Kovilpati-628503,Tamil Nadu,India

Reviewer #4: No

---

## [Author Response · Author response to Decision Letter 0]

13 Oct 2023

Journal Requirements:

We tried to modify the presentation of the paper according to the style rules. Hope that it meets your expectations.

Our code is uploaded along with the submission of the revised paper in editorial system.

We have not created any new dataset in this work, but we use the dataset introduced in Ref. [40] which is accessible from https://www.fhwa.dot.gov/publications/research/operations/07030/

4. Please ensure that you refer to Figure 2 in your text as, if accepted, production will need this reference to link the reader to the figure.

We do apologize for the mistake. The mentioned figure has been referred to in revised manuscript.

5. Please include a copy of Table 2 which you refer to in your text on page 15.

We do apologize for the mistake. The mentioned table has been referred to in revised manuscript.

Reviewer #1

As a reviewer, I read this article several times and found the topic very interesting and timely. Further, I found that this paper is well-written and organized and deals with a very interesting approach. However, below are some of the comments that I think the authors need to address before accepting the paper. The aim of this comment is to improve the quality of the paper.

1. The title is informative and relevant, but it sounds not interesting. I suggest rewriting the title to be more interesting.

Thank you for your recommendation. We modified the title from “Resource-Aware Cluster-based Routing…” to “Resource-Aware Re-Clustering…” to make it more informative.

2. The aim of the research is clear and the main article matches the abstract. However, it is not very clear what the study found. Therefore, I suggest writing the research result.

Thank you for this point. The achieved improvements have been explicitly mentioned in the abstract as highlighted in yellow.

3. I suggest writing the main contribution of the paper in the introduction section.

Thank you for this great point. We improved the writing of the Introduction section and explicitly mentioned the contributions in 6th paragraph of that section. We also add a section namely “Resource-aware routing in C-V2X” to the paper to better explain the contribution. 

4. It would be better to make a comparison table for related work and explain how the proposed research work differs from the existing one.

Thank you for this comment. Table 2 added in Related Work section to show the shortcomings of previous routing/resource allocation methods compared to proposed method.

5. The data results in Figure 5, Figure 6, and Figure 7 are relevant and presented clearly. However, it would be better to provide more analysis and discuss the results from multiple angles.

Thank you for this comment. We add Figure 8 to compare the methods in terms of the QoS support (the number of vehicles that their required rate is supported).

6. All related and recent references should be seen. For example, the following reference is closely related and I recommend citing them:

http://ijece.iaescore.com/index.php/IJECE/article/view/24016

https://online-journals.org/index.php/i-jim/article/view/26663

Thank you for the introduced references. The introduced references added and cited in Paragraph 5 of first section (Refs. [21] and [23]). Moreover, some other new references have been added and cited in revised manuscript (including Refs. [2], [4], [12], [22], [27], [28], and [39]). 

7. The conclusion answers the aims of the study and is supported by results, but it is missing future work. Therefore, it would be better to inform future research opportunities.

Thank you for this comment. Some suggested future works added to the Conclusion section.

Reviewer #2

Resource- Aware Cluster-based Routing in Hybrid C-V2X/DSRC Vehicular Networks

In this paper, the author proposes Resource aware Cluster-based Routing for C-V2X/DSRC Vehicular communication. The reviewer has several concerns as follows:

1. All abbreviations should be defined in the first place, eg., IEEE, 4G, LTE, 3GPP, UE, RSU, C-V2X..and so on. vehicle-to-vehicle should be V2V defined in Introduction. In many places abbreviation and acronyms are missed out. The same acronym in abstract and another part of the paper should be defined in abstract and another place in the first place.

Thank you for this comment. The text reviewed again and all acronyms defined in first place as requested. Also, Table 1 added for definition of all abbreviations used throughout the paper.

2. What are the main challenges in resource-aware cluster-based routing for vehicular networks? In order to appeal to interested readers, the authors need to point out the underlying difficulties in the studied work.

Thank you for this suggestion. Table 2 added in Section II to show the difficulties of previous routing/resource allocation methods regarding the aims of our work. We also add a section namely “Resource-aware routing in C-V2X” to the paper to better explain the challenge.

3. The study found that few existing algorithms consider negative effects induced by resource management in cellular networks for c-v2x communication. How to evaluate the rationality of the designed criteria considering the kinds of factors of ineffective HO in this study?

Thank you for your comment. Our method leads to some handovers between neighboring BSs due to cluster changes in the first time where the clusters are re-arranged after initialization. But in next iterations, the initial DBSCAN-based clustering is not performed and so the clusters are not changed significantly except due to mobility as is obvious from the flowchart of Fig. 5. We have mentioned this in paragraph before Fig. 5 (3rd paragraph of section entitled “Clarifying Example”). Studying the impact of proposed method on the number of handovers under various mobility patterns is mentioned as a future work. 

4. In this work how to ensure that the quality requirements are satisfied?

Thank you for this great point. We add Fig 8 to compare the methods in terms of the number of vehicles that their required cellular resources are provided and their needs are supported.

5. To provide enough and more information about various communication technologies and brief background about VANET, the following articles are suggested to be used and cited in place of ref 1, 2 in the Introduction. New References

[1] A two-phase fuzzy based access network selection scheme for vehicular ad hoc networks. Peer-to-Peer Networking and Applications. (https://link.springer.com/article/10.1007/s12083-021-01228-w)

[2] Two-phase access network selection scheme based on weighted sum and game theoretical approaches for vehicular ad hoc networks. Journal of Circuits, Systems and Computers, 30(11),2150206. (https://www.worldscientific.com/doi/abs/10.1142/S0218126621502066)

The following article is suggested to be cited instead of reference [11]

[11]. An efficient data transmission in VANET using clustering method. International Journal of Electronics and Telecommunications. (https://journals.pan.pl/dlibra/show-content?id=106023)

Thank you for introducing such references. We cited them in appropriate locations in revised manuscript (added as Refs. [2][4][16]).

6. Compared with the existing work, what are the key research gaps and contributions of this paper? The authors are suggested to clarify their contribution's specific research problem and novelty in the Introduction Section. Moreover, the reviewed related works in Section II should correspond to the research gaps. Explore the related works in more detail with 3 dimensions: Resource aware, Clustering and Routing. The related work should be re-analysed and reorganised. In related works under the theme routing, I suggest to include the below article instead of Ref. [15]

[15] A Fuzzy Based Trust And Priority Enabled AODV Routing Scheme For Vehicular Network. INTERNATIONAL JOURNAL OF SCIENTIFIC & TECHNOLOGY RESEARCH.

Thank you for your suggestions. We improved the writing of the Introduction section by explicitly enumerating the contributions in 6th paragraph of that section. Also, we add a section to the paper, namely “Resource-aware routing in C-V2X” section, to better explain the contribution. 

Moreover, we add Table 2 to 2nd section to compare the studied works and their shortcomings in brief. The related work is now better organized. Paragraph 2 of 2nd section is on resource allocation methods and Paragraph 3 is on routing and clustering methods.

The introduced reference has also been added and discussed in Paragraph 5 of first section as Ref. [22]. 

7. The proposed System model is vague for the reader. Including an illustrative figure for better understanding of the system scenario is highly suggested.

Writing of the “System model” subsection is improved to remove ambiguities. Also, Fig 3 is improved and some texts added in paragraph before Table 3 to better demonstrate the system model. 

8. The proposed model has 3 variants (3 Algorithms). Why are they necessary? How do they stand during experimental comparison/evaluation?

We do apologize for the ambiguity. The proposed method does not have 3 variants. The first algorithm is the basic DBSCAN-based vehicle clustering method that is executed only during initialization of clusters. Thereafter, Alg. 2 (and inside it, Alg. 3) are executed to update the clusters iteratively. We add the 3rd paragraph to the section entitled “Clarifying example” to explain this matter which is also obvious from the updated Fig. 5.

9. How does the proposed resource-aware cluster-based routing approach compare to existing methods in terms of performance and complexity?

Thank you for this suggestion. We added a sentence about the complexity of the proposed method and compared it to the basic DBSCAN algorithm in 3rd paragraph of the section entitled “Clarifying example”. 

10. The authors have well introduced the proposed work; however, the downside of this proposed work is not mentioned in detail. Authors should give a brief explanation related to the disadvantages of the proposed method, and also the main challenges (other than the computational complexity).

As we added in Conclusion section, the proposed method does not pay attention to the capacity of DSRC in supporting cluster members. Considering an analytical limit on the number of vehicles that DSRC can support in a cluster, and modifying the proposed method based on this limit, will be addressed as a future works. Also, the impact of the proposed method on the number of handovers should be studied as a drawback.

11. What do you mean by middle in [16] in Introduction section.

We do apologize for the ambiguity. The sentence “cluster-based routing protocols offer higher benefits in the middle [16]” corrected to “cluster-based routing protocols offer higher benefits in V2X [24]” in 5th Paragraph of Introduction section.

12. Why do you go for DBSCAN clustering mechanism when there are several other mechanisms?

Thank you for this comment. The 5th paragraph of Introduction section improved to show the motivations behind using DBSCAN. Also, in System model subsection, we emphasized that we have chosen DBSCAN “because, in contrast to the mean-shift algorithm [25], [38], scattered vehicles outside the vehicle density range are considered, it is not necessary to predefine the number of clusters, and it constructs clusters with unusual shapes in low time complexity [19]”.

13. Proper references should be provided for the equations that were borrowed from the literature.

All equations that are from previous work have been cited carefully. Others are proposed by the authors.

14. There are lot of figures without much necessity or explanation in the paper, e.g., Figure 1, 2 which makes the paper less readable. Remove the above figures. The authors are suggested to draw an illustrative figure about clustering in VANET scenario in Introduction section.

Thank you for your comment. We moved Fig 1 (Fig 2 in revised version) to a new section called “Resource-aware routing in C-V2X” and added more explanations to make the idea and our contribution clearer. Also, a new figure (Fig 1) added to this section for better illustration of the domain of the work. Also, Fig 2 (Fig 3 in revised version) improved and more discussed (in paragraph before Table 3) to remove the ambiguity.

15. The comparison between proposed method and traditional method in Figure 5, 6 is insufficient. What is the “traditional method” that is mentioned of? Make comparisons with existing other approaches to validate the results.

Thank you for this point. We add Fig 8 to compare the methods in terms of QoS support, i.e., the number of vehicles that their required cellular resources are provided and their needs are supported.

Also, the term “traditional method” replaced by Ref. [37] in figures. To the best of our knowledge, there is no other previous method that considers both clustering and RA to be compared to ours fairly. 

16. There remain many typos and grammatical errors. The writing of the paper needs to be improved from informal to formal way. (eg., A question may arise…), please reframe such sentences. Hence it is requested to proofread your text.

We do apologize for the writing of the paper. We do our best effort to review the paper thoroughly by the help of an expert and correct the writing typos and grammatical errors. To avoid complexity, the writing improvements have not been highlighted in the paper.

Reviewer #3 

Article ID: PONE-D-23-25645

Title: "Resource-Aware Cluster-based Routing in Hybrid C-V2X/DSRC Vehicular Networks

Comments:

1. It's interesting to note that DBSCAN, being a well-established clustering algorithm, has been part of the field for a while. The inclusion of re-clustering to form a new methodology adds a fresh perspective to its application. Considering your innovative approach, I wonder if there's room for exploring and producing new clustering concepts within this methodology. This could potentially push the boundaries even further and contribute to the advancement of the field such as grid based, ordering points based, hierarchy based etc…… It will improve the Quality of ur paper.

Thank you for this great point. As the suggested idea needs much work and evaluations to be expanded to other clustering algorithms, we just emphasized it in the Conclusion section that “employing proposed resource-awareness concept in other routing mechanisms and clustering algorithms can be attended in future works.”

2. The article quite informative and appreciate the coverage of the contents. However, it seems that there's a gap in the expansion of certain concepts like DRSC and LTE. Including these would provide a more comprehensive view and enhance the reader's (new) understanding. 

Thank you for this comment. Second paragraph of Introduction section is added to introduce mentioned technologies in more detail. 

3. The references and citations predominantly span up until the year 2015 (avoid 2007). To provide a more up-to-date context and incorporate recent advancements, I recommend considering references beyond that year. This would enhance the relevance and currentness of the article.

Thank you for this comment. The only old reference is last reference which cites the exploited dataset. As the exploited dataset is famous and widely used, we have exploited it for our evaluations. Other references are from 2015 and later. By the way, we added and cited some new references to improve its freshness, such as [2], [4], [12], [16], [21], [22], [23], [27], [28], and [39].

4. This article addresses the challenge of unbalanced load. Could you kindly provide more insight into how this is achieved through enhanced DBSCAN approach? Specifically, I'm interested in understanding the separate methodology you've developed to balance the load within networks. Elaborating on this aspect would greatly enhance the practical value of the research.

This is a great point. We add a paragraph before Eq ([Disp-formula pone.0293662.e012]) to discuss how the proposed algorithm improves the load balance of BSs.

5. the manuscript shows promise in addressing load balancing, clustering, and re-clustering. To enhance clarity, consider adopting a module-wise approach. Clearly outline each module's purpose, methodologies with equations, benefits, and practical cases with suitable diagrams. This will provide readers a systematic understanding. Looking forward to the refined manuscript.

Thank you for your suggestion. We improved Fig. 5 and the paragraph before it to show how the proposed algorithms work together. Each algorithm is invoked as an independent module in this flowchart.

6. The manuscript has highlighted important concerns associated with frequent reclustering, such as resource intensity, overfitting, maintenance challenges, and lack of stability. Addressing these challenges is essential for the practical application of frequent reclustering. Could you elaborate on potential strategies or methodologies that can be employed to overcome or mitigate these drawbacks? 

Thank you for this comment. Since overcoming your mentioned drawbacks needs a lot of work, we improved the Conclusion section to emphasize on these shortcomings and suggesting their solutions as future work.

7. In this article, eNodeB is not discussed except in the table. Could you provide a brief explanation of what eNodeB refers to and its importance?

We introduced eNodeB in 3rd paragraph of Section I.

Reviewer #4

The paper seems to be very timely and focuses on a topic that is very paramount as it relates to Vehicle to Everything (V2X) communication in 5G wireless networks. In V2X communications, the issue of resource availability, latency and stability have posed several challenges especially when it comes to IoVs. The topic presented is good, but this paper still suffers from major issues as given below:

1. The authors should improve on the literature review section of this manuscript. In section II, a table should be developed which compares the contributions of this work to other recent manuscripts in this field especially papers where 5G wireless networks have been used with IoVs technologies for wireless signal propagation or a similar scenario. The focus and coverage of the work, its limitations should also be included in the table.

Thank you for this comment. We added Table 2 to 2nd section of the paper to compare the studied works and their shortcomings in brief. The related work is now better organized. Paragraph 2 of Related Work section is on resource allocation methods and Paragraph 3 is on routing and clustering methods. Hope that the modifications meet your expectations. 

2. The abstract needs to be re-written and some sentences needs to be rephrased. For example, this sentence needs to be rephrased “It is while the routing protocols could help redirect the load of busy BSs to neighboring BSs if those protocols were resource-aware”

” Traditional cluster-based routings do not attend the resource availability of cellular BSs in their decisions” Please rephrase those sentences and many more so they make meaning.

We do apologize for the writing of the paper. We do our best effort to review the paper thoroughly by the help of an expert and correct the writing typos and grammatical errors. To avoid complexity, the writing improvements have not been highlighted in the paper.

3. There are many grammatical errors and incomplete sentences in the manuscript. The authors should correct those uncompleted sentences

We do apologize for the writing of the paper. We do our best effort to review the paper thoroughly by the help of an expert and correct the writing typos and grammatical errors.

4. In figure 1, Is this figure original or copied from somewhere? If copied, appropriate citation should be added. Same holds for Figure 2.

Both figures are original.

5. The contribution of this work needs to be well spelt out in the manuscript. List the contributions of this work at the end of the introduction in Section I.

Thank you for your suggestions. We improved the writing of the Introduction section by explicitly enumerating the contributions in 6th paragraph of that section. We also add a section namely “Resource-aware routing in C-V2X” to the paper to better explain the contribution. 

6. The methodology section of this work needs to be properly enhanced. In section III, the proposed model needs to be well developed. The authors should expand on this section and show all the stages of the model development. All variables for the model equations should be well defined and explained as it relates to this study.

Thank you for your consideration. We add more descriptions about some variables and equations in Proposed Method section. Also, Fig. 5 improved to better show the usage of each of the proposed algorithms in suggested solution.

7. Nowadays, the proliferation of massive wireless devices enables the transmission of sensitive user information over open communication channels. However, the security and privacy of such critical data have been worrisome. How do you guarantee the protection of sensitive data in IoVs communications? One would expect that this important aspect should be given significant consideration in the current work. What do you think?

Thanks for this comment. Security and privacy are important topics in VANET routing protocols and have been investigated in some previous works. However, the proposed method is on radio resource management during clustering and an existing clustering algorithm is extended accordingly. Therefore, the security and privacy is not in the scope of the paper and may be investigated as an independent work in future. 

8. In the conclusion section, a brief summary of the key findings from the work is requested. It is imperative to state the key takeaways from the work and their implications

This is a great point. We improved the Conclusion section to more emphasize on improvements and shortcomings of our method. 

9. List of abbreviations table should be included in this work. Some terms have been used without the authors defining those terms.

The text reviewed again and all acronyms defined in first place as requested. Also, Table 1 added for definition of all abbreviations used throughout the paper.

10. More authoritative references should be cited. The references need to be increased.

Thank you for this suggestion. Some new references added and cited in revised manuscript including Refs. [2], [4], [12], [16], [21], [22], [23], [27], [28], and [39].

---

## [Decision Letter · Decision Letter 1]

16 Oct 2023

Resource-Aware DBSCAN-based Re-Clustering in Hybrid C-V2X/DSRC Vehicular Networks

PONE-D-23-25645R1

Dear Dr. Shahgholi Ghahfarokhi,

We’re pleased to inform you that your manuscript has been judged scientifically suitable for publication and will be formally accepted for publication once it meets all outstanding technical requirements.

Kind regards,

C. Suganthi Evangeline, B.E.,M.E.,Ph.D

Academic Editor

PLOS ONE

Additional Editor Comments (optional):

Reviewers' comments:

Reviewer's Responses to Questions

**Comments to the Author**

1. If the authors have adequately addressed your comments raised in a previous round of review and you feel that this manuscript is now acceptable for publication, you may indicate that here to bypass the “Comments to the Author” section, enter your conflict of interest statement in the “Confidential to Editor” section, and submit your "Accept" recommendation.

Reviewer #2: All comments have been addressed

Reviewer #3: All comments have been addressed

2. Is the manuscript technically sound, and do the data support the conclusions?

Reviewer #2: Yes

Reviewer #3: Yes

3. Has the statistical analysis been performed appropriately and rigorously? 

Reviewer #2: N/A

Reviewer #3: Yes

4. Have the authors made all data underlying the findings in their manuscript fully available?

Reviewer #2: Yes

Reviewer #3: Yes

5. Is the manuscript presented in an intelligible fashion and written in standard English?

Reviewer #2: Yes

Reviewer #3: Yes

6. Review Comments to the Author

Reviewer #2: (No Response)

Reviewer #3: The author has considered and addressed the provided comments, and the editor may proceed with acceptance.

7. PLOS authors have the option to publish the peer review history of their article (what does this mean?). If published, this will include your full peer review and any attached files.

Reviewer #2: No

Reviewer #3: No

---

## [Editor Report · Acceptance letter]

20 Oct 2023

PONE-D-23-25645R1 

Resource-Aware DBSCAN-based Re-Clustering in Hybrid C-V2X/DSRC Vehicular Networks 

Dear Dr. Shahgholi Ghahfarokhi:

I'm pleased to inform you that your manuscript has been deemed suitable for publication in PLOS ONE. Congratulations! Your manuscript is now with our production department. 

Kind regards, 

on behalf of

Dr. C. Suganthi Evangeline 

Academic Editor

PLOS ONE